# FAM83F regulates canonical Wnt signalling through an interaction with CK1α

Karen Dunbar[1], Rebecca A Jones[2], Kevin Dingwell[2], Thomas J Macartney[1], James C Smith[2], Gopal P Sapkota[1]

The function of the FAM83F protein, like the functions of many members of the FAM83 family, is poorly understood. Here, we show that injection of Fam83f mRNA into *Xenopus* embryos causes axis duplication, a phenotype indicative of enhanced Wnt signalling. Consistent with this, overexpression of FAM83F activates Wnt signalling, whereas ablation of FAM83F from human colorectal cancer (CRC) cells attenuates it. We demonstrate that FAM83F is farnesylated and interacts and co-localises with CK1α at the plasma membrane. This interaction with CK1α is essential for FAM83F to activate Wnt signalling, and FAM83F mutants that do not interact with CK1α fail to induce axis duplication in *Xenopus* embryos and to activate Wnt signalling in cells. FAM83F acts upstream of GSK-3β because the attenuation of Wnt signalling caused by loss of FAM83F can be rescued by GSK-3 inhibition. Introduction of a farnesyl-deficient mutant of FAM83F in cells through CRISPR/Cas9 genome editing redirects the FAM83F–CK1α complex away from the plasma membrane and significantly attenuates Wnt signalling, indicating that FAM83F exerts its effects on Wnt signalling at the plasma membrane.

## Introduction

FAM83F belongs to the FAM83 family of proteins, which is characterised by a conserved N-terminal DUF1669 domain. We have recently shown that the DUF1669 domain mediates interaction with the α, δ, or ε isoforms of the CK1 family of Ser/Thr protein kinases (1). The FAM83 proteins direct the CK1 isoforms with which they interact to distinct subcellular compartments, thereby potentially regulating their substrate pools (1). All FAM83 proteins interact with CK1α, albeit with varying affinity, while FAM83A, B, E, and H also interact with CK1δ and ε isoforms (1). CK1α, δ and ε isoforms have been implicated in numerous cellular processes including Wnt signalling, mitosis, circadian rhythm, and DNA damage responses (2, 3, 4, 5).

There is now increasing evidence that FAM83 proteins regulate the diverse biological roles of CK1 isoforms. For example, FAM83G

(also known as PAWS1) regulates canonical Wnt signalling downstream of the β-catenin destruction complex through association with CK1α (6). Interestingly, two mutations within the DUF1669 domain of the *FAM83G* gene that cause palmoplantar keratoderma result in the loss of FAM83G-CK1α interaction and attenuation of Wnt signalling (7). FAM83D directs CK1α to the mitotic spindle to ensure proper spindle alignment and timely exit from mitosis (8), and FAM83H mutations that cause amelogenesis imperfecta retain interaction with CK1 isoforms but are mis-localised in cells (9, 10). However, the biological and biochemical roles of FAM83F are poorly understood. High levels of FAM83F protein have been linked to oncogenesis in glioma (11), lung cancer (12), oesophageal cancer (13), and thyroid cancer (14) yet the underlying mechanisms remain unknown. Sequence alignment of the conserved DUF1669 domain reveals that FAM83F most resembles FAM83G and is the only other FAM83 protein to induce Wnt reporter activity in an overexpression assay (Fig S1). We, therefore, sought to explore whether FAM83F is also involved in regulating canonical Wnt signalling.

Wnt signalling plays important roles in embryogenesis and cell proliferation as well as in stem cell and adult tissue homeostasis (15). The key effector of the canonical Wnt signalling pathway is β-catenin. Under basal conditions, most β-catenin protein is located at the adherens junctions, whereas cytoplasmic β-catenin levels are kept in check by the β-catenin destruction complex. The destruction complex is composed principally of two scaffold proteins, Axin and Adenomatous polyposis coli (Apc), and two protein kinases, glycogen synthase kinase-3β (GSK-3β), and casein kinase 1α (CK1α). Phosphorylation of β-catenin at S45 by CK1α primes β-catenin for GSK-3β–mediated sequential phosphorylation at T41, S37, and S33, which allows the β-transducin repeat-containing protein (β-Trcp) to ubiquitylate β-catenin and facilitate its degradation through the proteasome (16). Upon binding Wnt ligands, the Wnt receptor Frizzled and co-receptor LRP6 recruit Dishevelled and the β-catenin destruction complex to the plasma membrane. This complex, termed the Wnt signalosome, in turn becomes internalised in multivesicular bodies, thus sparing the degradation of cytoplasmic β-catenin (17). The resultant stabilised β-catenin then translocates to the nucleus, where it binds to its co-transcriptional factor, T-cell factor (TCF), and triggers the transcription of Wnt target

[1]Medical Research Council Protein Phosphorylation and Ubiquitylation Unit (MRC-PPU), School of Life Sciences, University of Dundee, Sir James Black Centre, Dundee, UK
[2]The Francis Crick Institute, London, UK

Correspondence: g.sapkota@dundee.ac.uk

genes, such as *Axin2*, *C-myc*, and *Cyclin D1* [18]. Aberrant Wnt signalling is a common feature in various cancers, particularly those of gastrointestinal origin including a vast majority of colorectal cancers (CRC) [19].

In this study we explore the role of FAM83F in driving Wnt signalling in *Xenopus* embryos and tissue culture cells, including CRC cells.

# Results

## FAM83F induces axis duplication in *Xenopus* embryos through an interaction with CK1α

The activation of the canonical Wnt signalling pathway by ectopic expression of Wnt ligands and mediators in early *Xenopus* embryos causes axis duplication [20]. Previously, we showed that injection of *Xenopus* embryos with *FAM83G* mRNA into a ventral blastomere at the four-cell stage induced secondary axis formation [6]. Ectopically expressing mRNA in early *Xenopus* embryos is thus an efficient method for screening potential regulators of canonical Wnt signalling. Upon injection of axis-inducing mRNA, four possible axial phenotypes can result, including complete secondary axes, partial secondary axes, dorsalised embryos, and those resembling wild-type (Fig 1A). To test the impact of FAM83F on *Xenopus* embryos, 500 pg of mRNA encoding HA-tagged zebrafish fam83fa, which closely resembles FAM83F in human and other species and was the only construct available at the time, was injected into a single ventral blastomere at the four-cell stage. Embryos were maintained until approximately stage 35, at which point we counted the embryos displaying each class of axial phenotype. *HA-fam83fa* induced secondary axes in >60% of the *Xenopus* embryos (Fig 1B–D).

FAM83F interacts selectively with CK1α through its conserved DUF1669 domain and mutating the phenylalanine residues from a conserved F-X-X-X-F motif to alanine abolishes this interaction [1]. In zebrafish Fam83fa, these two phenylalanine residues map at amino acid positions 275 and 279. Mutation of either F275 or F279 to an alanine prevented the induction of a secondary axis (Fig 1B and C), as did the double mutant, Fam83fa$^{F275A/F279A}$, despite all proteins being expressed, as shown by Western blot (Fig 1D). This indicates that Fam83fa-CK1α binding is required for Fam83fa to induce axis duplication in *Xenopus* embryos. When we tested Fam83fa fragments with deletions at the C terminus for their ability to induce axis duplication, both full-length *fam83fa* mRNA (*HA-fam83fa$^{1-555aa}$*) and the DUF1669 domain fragment (*HA-fam83fa$^{1-300aa}$*) induced secondary axes robustly (Fig S2A–C). Interestingly, the C-terminal deletions *HA-fam83fa$^{1-500aa}$*, *HA-fam83fa$^{1-400aa}$*, and *HA-fam83fa$^{1-356aa}$* induced secondary axes poorly, indicating that loss of the C-terminal portion of the protein affects Fam83fa structure or function.

Human FAM83F contains a protein prenylation motif, a conserved CAAX box sequence at the C terminus in which the cysteine residue is modified through an addition of either a geranylgeranyl or a farnesyl moiety [21]. When FAM83F that has been overexpressed in HEK-293 cells was isolated, cleaved with trypsin and subjected to mass-spectrometry, we identified tryptic peptides

found to be farnesylated at Cys497 (Fig S3A–C). Farnesylation, a posttranslational modification, involves the addition of a 15-carbon farnesyl group to a C-terminal cysteine residue by farnesyltransferase; this plays a role in the regulation of protein-membrane interactions and in signal transduction circuits [22]. Zebrafish Fam83fa, which possesses the CIQS sequence at its C terminus, is also predicted to be farnesylated [23]. Mutation in the human protein of the CAAX-box invariant cysteine to an alanine, creating FAM83F$^{C497A}$, prevents farnesylation but injection of mRNA encoding this mutated protein induced secondary axis formation in *Xenopus* embryos in a similar manner to that of wild-type *FAM83F* (Fig S2D). This indicates that farnesylation of the C terminus of FAM83F is not required for its ability to activate canonical Wnt signalling in the *Xenopus* assays.

## FAM83F-CK1α interaction is required for FAM83F membrane localisation and canonical Wnt signalling effects

Initial investigations of FAM83F biology were performed in the U2OS Flp-In T-Rex (Flp/Trx) cell line, which allows doxycycline inducible gene expression of stably integrated proteins of interest. Stable cell lines were generated to express GFP only, GFP-FAM83A, GFP-FAM83F, GFP-FAM83F$^{C497A}$, GFP-FAM83F$^{D250A}$, and GFP-FAM83F$^{F284A/F288A}$ (Fig 2A). In addition to the previously described conserved F-X-X-X-F motif, which is required for the FAM83-CK1 interactions, a separate conserved residue that maps to an aspartic acid at 250 in FAM83F was identified which when mutated to an alanine can also disrupt FAM83-CK1 interactions [1] (Fig 2A). Immunoprecipitation of GFP confirmed that GFP-FAM83F interacts with CK1α but no interaction was detected with GFP only or with GFP-FAM83A (Fig 2B). GFP-FAM83F$^{F284A/F288A}$ does not interact with CK1α, while the interaction between GFP-FAM83F$^{D250A}$ and CK1α is severely reduced compared with wild type GFP-FAM83F (Fig 2B). The farnesyl-deficient mutant, GFP-FAM83F$^{C497A}$, still maintains an interaction with CK1α (Fig 2B).

Fluorescence microscopy shows that GFP-FAM83F is expressed at both the plasma membrane and nucleus in U2OS Flp/Trx cells (Fig 2C). The farnesyl-deficient mutant GFP-FAM83F$^{C497A}$ was detectable only in the nucleus, suggesting that farnesylation of FAM83F directs its localisation to the plasma membrane. Interestingly, the two CK1α-binding deficient mutants, GFP-FAM83F$^{D250A}$ and GFP-FAM83F$^{F284A/F288A}$, exhibited cytoplasmic and peri-nuclear localisation away from the plasma membrane and the nucleus. This suggests that the membrane and nuclear localisation of FAM83F is facilitated by its association with CK1α. Co-staining with an anti-CK1α antibody revealed overlapping localisation with GFP-FAM83F and GFP-FAM83F$^{C497A}$, but not with GFP-FAM83F$^{D250A}$ and GFP-FAM83F$^{F284A/F288A}$ (Fig 2C).

Canonical Wnt signalling activity can be measured using a dual luciferase reporter assay in which cells are transfected with a plasmid containing either wild-type TCF, a co-transcriptional activator of β-catenin, binding sites (TOPflash) or mutant TCF-binding sites (FOPflash) upstream of a luciferase reporter [24]. An increase in "free" β-catenin protein, following activation of Wnt signalling, interacts with the TCF-binding sites, thereby inducing luciferase expression and hence activity. Overexpression of GFP-FAM83F and GFP-FAM83F$^{C497A}$ significantly increased luciferase reporter activity in cells treated with control L-conditioned medium compared with

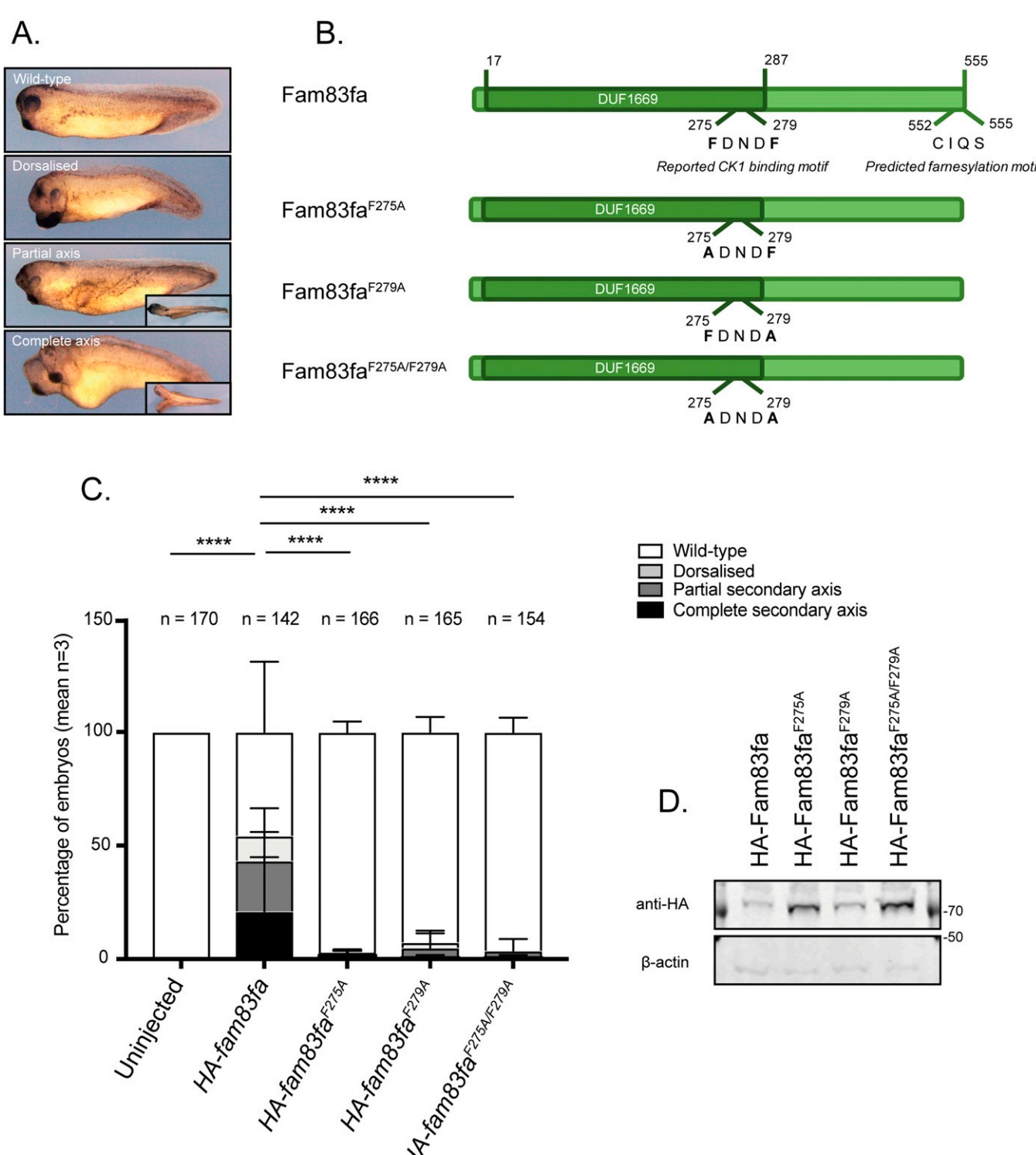

**Figure 1. Fam83f induces axis duplication in *Xenopus* embryos through an interaction with casein kinase 1α (CK1α).**
**(A)** Bright-field microscopy images of *Xenopus* embryo axis duplication phenotypes; wild-type, dorsalised, partial secondary axis and complete secondary axis. **(B)** Cartoon of Fam83fa protein illustrating the location of the CK1 binding and farnesylation motifs and cartoons of the point mutants expressed in (C, D). **(C)** Percentage of *Xenopus* embryos showing phenotypes depicted in (A) following injection with HA-tagged zebrafish *fam83fa*, *fam83fa^F275A*, *fam83fa^F279A*, or *fam83fa^F275A/F279A* mRNA. Data represent three independent experiments with total numbers of *Xenopus* embryos denoted above the graph. Bar graph representing mean + SD. Statistical significance determined by two-way ANOVA with Tukey's post hoc test to compare the percentage of embryos displaying a wild-type phenotype. ****$P \leq 0.0001$. **(D)** Protein extracts from *Xenopus* embryos following injection with HA tagged zebrafish *fam83fa*, *fam83fa^F275A*, *fam83fa^F279A*, or *fam83fa^F275A/F279A* mRNA were resolved by SDS–PAGE and subjected to Western blotting with indicated antibodies.
Source data are available for this figure.

GFP controls (Fig 2D). In contrast, overexpression of the CK1α-binding deficient mutants GFP-FAM83F$^{D250A}$ and GFP-FAM83F$^{F284A/F288A}$ did not increase luciferase activity under these conditions. Following addition of Wnt3A-conditioned medium, GFP-FAM83F and GFP-FAM83F$^{C497A}$ cell lines had significantly increased luciferase activity when compared with Wnt3A-treated GFP controls (Fig 2D). Wnt3A-induced luciferase reporter activity in cells expressing the CK1α-binding deficient mutants GFP-FAM83F$^{D250A}$ and GFP-FAM83F$^{F284A/F288A}$ was substantially lower than in cells expressing GFP-FAM83F and GFP-FAM83F$^{C497A}$ (Fig 2D). This suggests that FAM83F-induced Wnt reporter activity is mediated through its association with CK1α. Expression of GFP tagged FAM83F proteins do not appear to affect the overall nuclear translocation of endogenous β-catenin basally or following Wnt3A stimulus, as observed by immunofluorescence and in cytoplasmic/nuclear fractions (Fig S4A and B). However, expression of GFP-FAM83F and GFP-FAM83F$^{C497A}$ increases cytoplasmic β-catenin levels basally compared with cells expressing GFP only or GFP-FAM83F$^{F284A/F288A}$, as demonstrated by subcellular fractionation (Fig S5). This increase in "free" β-catenin potentially explains the increased TCF-luciferase activity detected upon expression of GFP-FAM83F and GFP-FAM83F$^{C497A}$ as the dual-luciferase assay only requires β-catenin to bind to the artificial TCF promoter to induce luciferase activity.

### Endogenous FAM83F localises to the plasma membrane and interacts with CK1α

To facilitate the study of endogenous FAM83F protein we screened multiple tissues and cell lines to identify cell lines with detectable levels of endogenous FAM83F protein. Tissue-specific expression of FAM83F from mouse tissue extracts revealed that FAM83F protein was detected in spleen, lung, and gastrointestinal tissues, with the highest levels of FAM83F protein detected in the stomach, small intestine, large intestine, and intestinal crypts (Fig 3A). Similar assessment of a panel of routinely studied cell lines showed that FAM83F protein was detected in extracts from the mammary adenocarcinoma cell line MDA-MB-468, and the colorectal cancer cell line HCT116, but was undetectable in many other cell lines (Fig S6A and B). Separately, we detected FAM83F protein in extracts from HaCaT keratinocytes as well as DLD-1 colorectal cells (Fig 3A and B). Based on the abundance of FAM83F protein observed in gastrointestinal tissue extracts and colorectal cells, we proceeded with two colorectal cancer cell lines, HCT116 and DLD-1, for further investigation into the role of endogenous FAM83F in canonical Wnt signalling. By using CRISPR/Cas9 technology, we generated FAM83F knockout (FAM83F$^{-/-}$) HCT116 (clones 1 and 2) and DLD-1 cells and also knocked in a GFP tag N-terminal to the *FAM83F* gene homozygously in both cell lines (HCT116 $^{GFP/GFP}$FAM83F and DLD-1 $^{GFP/GFP}$FAM83F) (Fig 3B). Both knockouts and GFP-knockins were verified by DNA sequencing and Western blotting (Figs 3B and S7A and B).

GFP-FAM83F immunoprecipitates (IPs) from HCT116 $^{GFP/GFP}$FAM83F and DLD-1 $^{GFP/GFP}$FAM83F cell extracts but not from wild-type cells co-precipitated endogenous CK1α, but not CK1δ or CK1ε (Fig 3C). Similarly, endogenous CK1α IPs co-precipitated FAM83F from wild-type HCT116 and DLD-1 cell extracts but not from HCT116 FAM83F$^{-/-}$ (cl.1) and DLD-1 FAM83F$^{-/-}$ cell extracts (Fig 3D). The interaction between CK1α and

FAM83G was unaffected by FAM83F knockout because CK1α IPs co-precipitated FAM83G from all cell extracts (Fig 3D). Endogenously driven GFP-FAM83F in HCT116 $^{GFP/GFP}$FAM83F knock-in cells localised predominantly to the plasma membrane, as revealed by anti-GFP immunofluorescence (Fig 3E). Under these conditions, endogenous CK1α immunostaining exhibited a diffuse pan-cellular staining, which limited the detection of plasma membrane-specific co-localisation with GFP-FAM83F. Immunofluorescence using a FAM83F antibody in HCT116 wild-type cells confirmed the plasma membrane localisation of endogenous FAM83F, which co-localised with β-catenin, a marker of the adherens junctions (Fig 3F). The specificity of anti-FAM83F staining was demonstrated using HaCaT wild-type and FAM83F$^{-/-}$ cell lines (Fig S8).

### Knockout of FAM83F reduces canonical Wnt signalling in colorectal cancer cells

Constitutively active Wnt signalling is a hallmark of many colorectal cancers (CRCs) (19). Mutations in the *Apc* gene, which encodes a central component of the β-catenin destruction complex and facilitates the destruction of free cytoplasmic β-catenin, are among the most common in CRCs (25). DLD-1 cells express Apc truncated at residue 1417, whilst HCT116 cells have a heterozygous mutation of β-catenin at S45, which prevents β-catenin from being phosphorylated and then ubiquitinated and degraded (26, 27). Thus, constitutive activation of Wnt signalling in HCT116 and DLD-1 cells occurs through perturbations at different stages of the pathway, and the responses to Wnt3A-ligand stimulation are also likely to differ. Canonical Wnt signalling activity can be measured by the transcript expression of Wnt target genes such as *Axin2* (18) as demonstrated by the >2-fold increase in *Axin2* mRNA expression following the addition of Wnt3A-CM to wild-type HCT116 cells (Fig 4A). In both HCT116 FAM83F$^{-/-}$ clones, the Wnt3A-induced increase in *Axin2* mRNA abundance was significantly reduced compared with wild-type HCT116 cells. Neither wild-type DLD-1 cells nor DLD-1 FAM83F$^{-/-}$ cells were responsive to treatment with Wnt3A-CM, but DLD-1 FAM83F$^{-/-}$ cells had a slight but significant reduction in basal *Axin2* mRNA abundance. In addition to the two colorectal cancer cell lines, we generated a FAM83F knockout in the osteosarcoma cell line U2OS (Fig S9A). Endogenous FAM83F protein abundance is low in U2OS cells, with detection only possible after immunoprecipitation with an anti-FAM83F antibody. Canonical Wnt activity was determined using the dual luciferase assay (Fig S9B) and the endogenous Wnt target gene *Axin2* (Fig S9C). U2OS wild-type and U2OS FAM83F$^{-/-}$ cells respond to Wnt3A stimulus with increased luciferase activity and *Axin2* mRNA abundance, respectively, but the extent of both responses was significantly reduced in U2OS FAM83F$^{-/-}$ cells compared with U2OS wild-type cells (Fig S9B and C).

We next sought to determine whether the FAM83F-CK1α interaction is affected by Wnt3A stimulation. CK1α IPs co-precipitated endogenous FAM83F from wild-type HCT116 and DLD-1 cell extracts but not from HCT116-FAM83F$^{-/-}$ (cl.1) and DLD-1-FAM83F$^{-/-}$ cell extracts regardless of stimulation with L-CM or Wnt3A-CM for 6 h (Fig 4B). We also analysed the components of the canonical Wnt3A signalling pathway in these cells upon stimulation with Wnt3A-CM (Fig 4C). A slight reduction in phospho-β-catenin (S45) was detected

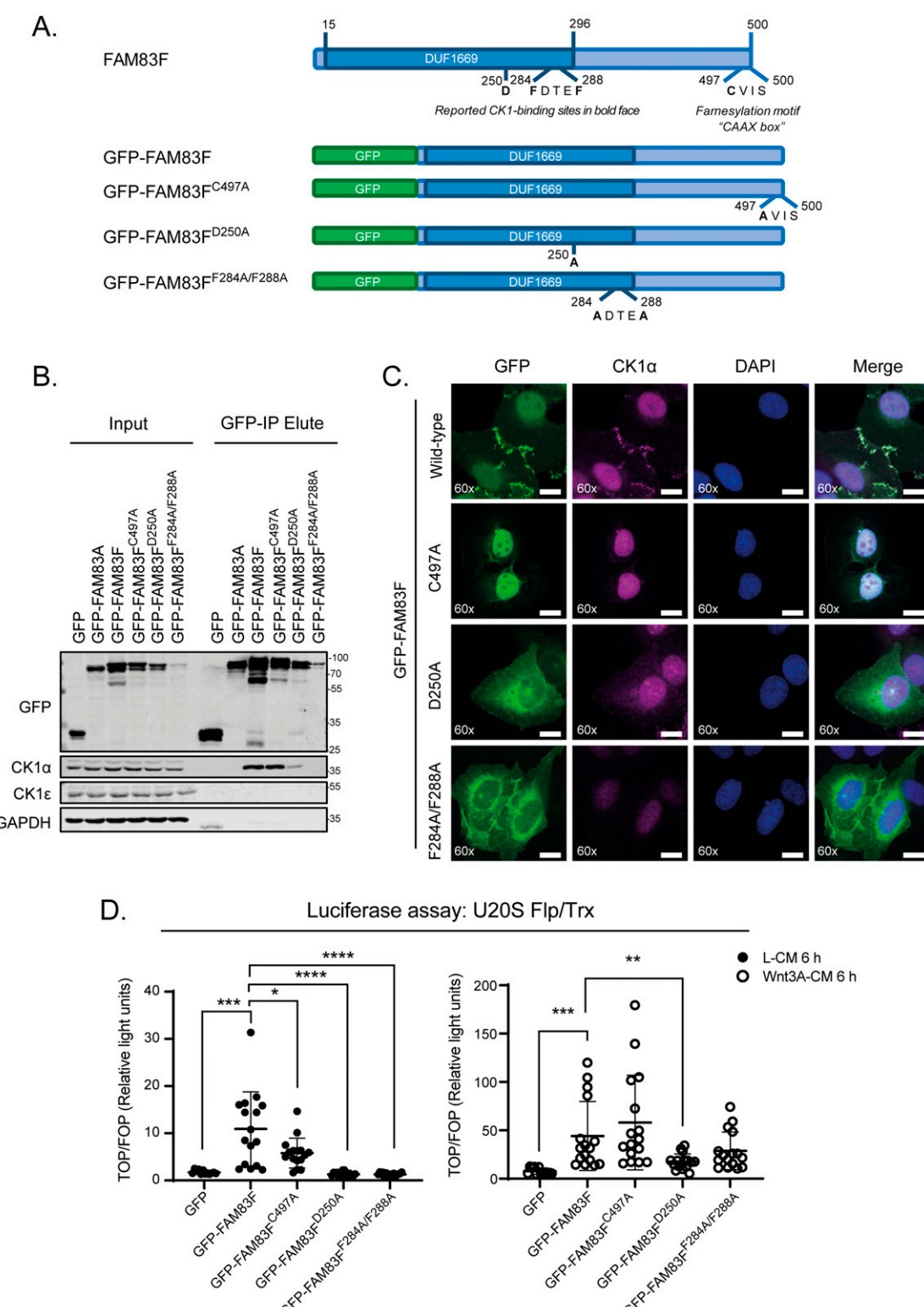

**Figure 2. FAM83F–casein kinase 1α (CK1α) interaction is required for membrane localisation and canonical Wnt signalling effects.**
**(A)** Cartoon of FAM83F protein illustrating the location of the CK1 binding and farnesylation motifs. Cartoons of the GFP-tagged FAM83F point mutants expressed in (B, C, D) are also indicated. **(B)** Lysates from U2OS Flp/Trx cells expressing GFP, GFP-FAM83A, GFP-FAM83F, GFP-FAM83F[C497A], GFP-FAM83F[D250A], or GFP-FAM83F[F284A/F288A] were subjected to immunoprecipitation with GFP trap beads. Input lysates and GFP IPs were resolved by SDS–PAGE and subjected to Western blotting with the indicated antibodies. **(C)** Representative wide-field immunofluorescence microscopy images of U2OS Flp/Trx cells expressing GFP-FAM83F, GFP-FAM83F[C497A], GFP-FAM83F[D250A], or GFP-FAM83F[F284A/F288A], labelled with antibodies recognising GFP (far left panels, green), CK1α (second row of panels from left, magenta), and DAPI (third row of panels from

in wild-type and FAM83F$^{-/-}$ HCT116 cells following addition of Wnt3A-CM compared with L-CM controls, indicating activation of the signalling pathway. However, the absence of phosphorylation of LRP6 and increase in active $\beta$-catenin following Wnt3A stimulus is surprising and may reflect the transient nature of phosphorylation within signalling pathways, or a poor response to Wnt3A stimulus due to constitutively active Wnt signalling in these cell lines. The levels of phospho-LRP6 (S1490), total LRP6, phospho-$\beta$-catenin (S33/S37/T41), active $\beta$-catenin, and total $\beta$-catenin were similar in HCT116 wild-type, HCT116 FAM83F$^{-/-}$ (cl.1) and HCT116 FAM83F$^{-/-}$ (cl.2) cells, regardless of Wnt3A stimulation (Fig 4C). Although there was a slight increase in LRP6 levels in FAM83F$^{-/-}$ DLD-1 cells relative to wild-type DLD-1 cells, no consistent changes in the abundance of any other Wnt signalling pathway proteins or phospho-proteins was evident between wild-type and FAM83F knockout DLD-1 cells, regardless of Wnt3A stimulation.

### FAM83F acts upstream of GSK-3$\beta$ and the loss of FAM83F protein reduces CK1$\alpha$ protein abundance at the plasma membrane

We sought to investigate where within the canonical Wnt/$\beta$-catenin signalling pathway FAM83F was acting. If the inhibition of Wnt signalling upon FAM83F loss can be restored by GSK-3 inhibitors, which prevent the phosphorylation and subsequent ubiquitin-mediated proteasomal degradation of $\beta$-catenin, this would imply that FAM83F acts upstream of GSK-3$\beta$. As previously noted, Wnt3A-induced *Axin2* mRNA abundance was lower in HCT116 FAM83F$^{-/-}$ cells than in wild-type HCT116 cells (Fig 4A). However, treatment of wild-type HCT116 and HCT116 FAM83F$^{-/-}$ cells with 0.5 $\mu$M CHIR99021, a selective GSK-3 inhibitor, enhanced *Axin2* mRNA abundance to a similar extent in both cell lines regardless of Wnt3A stimulation (Fig 5A). In a similar assay, when U2OS wild-type and U2OS FAM83F$^{-/-}$ cells were treated with 0.5 $\mu$M of CHIR99021, *Axin2* mRNA abundance increased to a similar extent in both cell lines regardless of Wnt3A stimulation (Fig S9C). These observations indicate that FAM83F acts to modulate Wnt signalling upstream of GSK-3$\beta$.

Given the plasma membrane localisation of FAM83F, as detected by immunofluorescence (Figs 2C and 3E and F), we hypothesised that FAM83F may influence canonical Wnt signalling at the membrane. Wild-type HCT116, HCT116 FAM83F$^{-/-}$ (cl.1), and HCT116 FAM83F$^{-/-}$ (cl.2) cells were separated into cytoplasmic, nuclear and membrane fractions (Fig 5B). In wild-type HCT116 cells, FAM83F was detected predominately in the membrane fraction with a small proportion in the nuclear fraction, whereas CK1$\alpha$ was detected in all three fractions: cytoplasmic, nuclear, and membrane (Fig 5B). Interestingly, CK1$\alpha$ protein abundance in the membrane fraction was significantly reduced in both HCT116 FAM83F$^{-/-}$ (cl.1) and HCT116

FAM83F$^{-/-}$ (cl.2) cells compared with HCT116 wild-type cells (Fig 5C). A similar result was observed using DLD-1 FAM83F$^{-/-}$ and DLD-1 wild-type cells (Fig 5D and E). These results suggest that FAM83F directs CK1$\alpha$ to the plasma membrane.

### Membranous localisation of FAM83F is required for FAM83F's role in canonical Wnt signalling

To investigate the importance of membrane bound FAM83F in canonical Wnt signalling under physiological conditions, we sought to knock-in a point mutation into the farnesylation motif of FAM83F, C497A, which should prevent membrane anchorage of FAM83F, using CRISPR/Cas9 genome editing (Fig 6A). We isolated three clones, HCT116 FAM83F$^{C497A}$ (clones 1–3) which were confirmed by genomic sequencing as homozygous (clones 1 & 3) and hetero-zygous (clone 2; the second allele had additional random insertions thereby also predictive of disrupting FAM83F membrane local-isation) for the FAM83F$^{C497A}$ knock-in (Fig S10A and B). Separation of HCT116 wild-type and FAM83F$^{C497A}$ cell extracts into cytoplasmic, nuclear, and membrane fractions show a significant reduction in FAM83F and CK1$\alpha$ protein in the membrane fractions in all HCT116 FAM83F$^{C497A}$ clones compared with wild-type cells (Fig 6B and C). CK1$\alpha$ IPs from HCT116 wild-type and HCT116 FAM83F$^{C497A}$ cell extracts co-precipitated FAM83F but not from HCT116 FAM83F$^{-/-}$ cells (Fig 6D), confirming that farnesylation of FAM83F does not affect CK1$\alpha$ binding. The role of FAM83F farnesylation in Wnt signalling was determined by evaluating *Axin2* mRNA expression in HCT116 wild-type and FAM83F$^{C497A}$ clones following treatment with L-CM or Wnt3A-CM for 6 h (Fig 6E). HCT116 FAM83F$^{C497A}$ clone 1 has signifi-cantly reduced *Axin2* transcript levels following Wnt3A stimulus compared to wild-type cells, but no significant changes were ob-served following L-CM treatment. HCT116 FAM83F$^{C497A}$ clones 2 and 3 have significantly reduced *Axin2* transcript levels compared to wild-type cells following both L-CM and Wnt3A-CM treatments. These results indicate that farnesylation of FAM83F is required for ca-nonical Wnt signalling in HCT116 cells.

Interestingly, the Wnt signalling deficit observed in all endog-enous FAM83F$^{C497A}$ clones is in stark contrast to the induction of Wnt signalling following overexpression of GFP-FAM83F$^{C497A}$ (Fig 2D). The overexpression of FAM83F proteins has been shown to alter the localisation of CK1$\alpha$ with GFP-FAM83F$^{C497A}$ increasing nuclear CK1$\alpha$ staining (Fig 2C). This re-direction of CK1$\alpha$ by GFP-FAM83F$^{C497A}$ reduces cytoplasmic CK1$\alpha$ levels (Fig S11A). In turn, cells over-expressing GFP-FAM83F$^{C497A}$ have significantly reduced levels of phosphorylated $\beta$-catenin at serine 45, which requires CK1$\alpha$ and triggers $\beta$-catenin ubiquitylation and degradation (Fig S11B and C). Therefore, it is not unexpected that cells overexpressing GFP-FAM83F$^{C497A}$ would be predicted to have increased "free"

---

left, blue). Overlay of GFP, CK1$\alpha$, and DAPI images as a merge is shown on the right. Immunofluorescence images captured with a 60× objective. Scale bar represents 10 $\mu$m. **(D)** Relative luciferase activity in U2OS Flp/Trx cells expressing GFP, GFP-FAM83F, GFP-FAM83F$^{C497A}$, GFP-FAM83F$^{D250A}$, or GFP-FAM83F$^{F284A/F288A}$ treated with either L- or Wnt3A-conditioned medium for 6 h. Luciferase activity is presented as TOPflash luciferase normalised to FOPflash luciferase and Renilla expression, the transfection control plasmid. Data presented as scatter graph illustrating individual data points with an overlay of the mean ± SD. Expression of GFP, GFP-FAM83A, GFP-FAM83F, GFP-FAM83F$^{C497A}$, GFP-FAM83F$^{D250A}$, and GFP-FAM83F$^{F284A/F288A}$ in U2OS Flp/Trx cells was induced by a treatment with 20 ng/ml doxycycline for 24 h. Statistical analysis of (C) was completed using a Student's unpaired *t* test and comparing cell lines as denoted on graph. Statistically significant *P*-values are denoted by asterisks (**** < 0.0001, *** < 0.001, ** < 0.01, * < 0.05).
Source data are available for this figure.

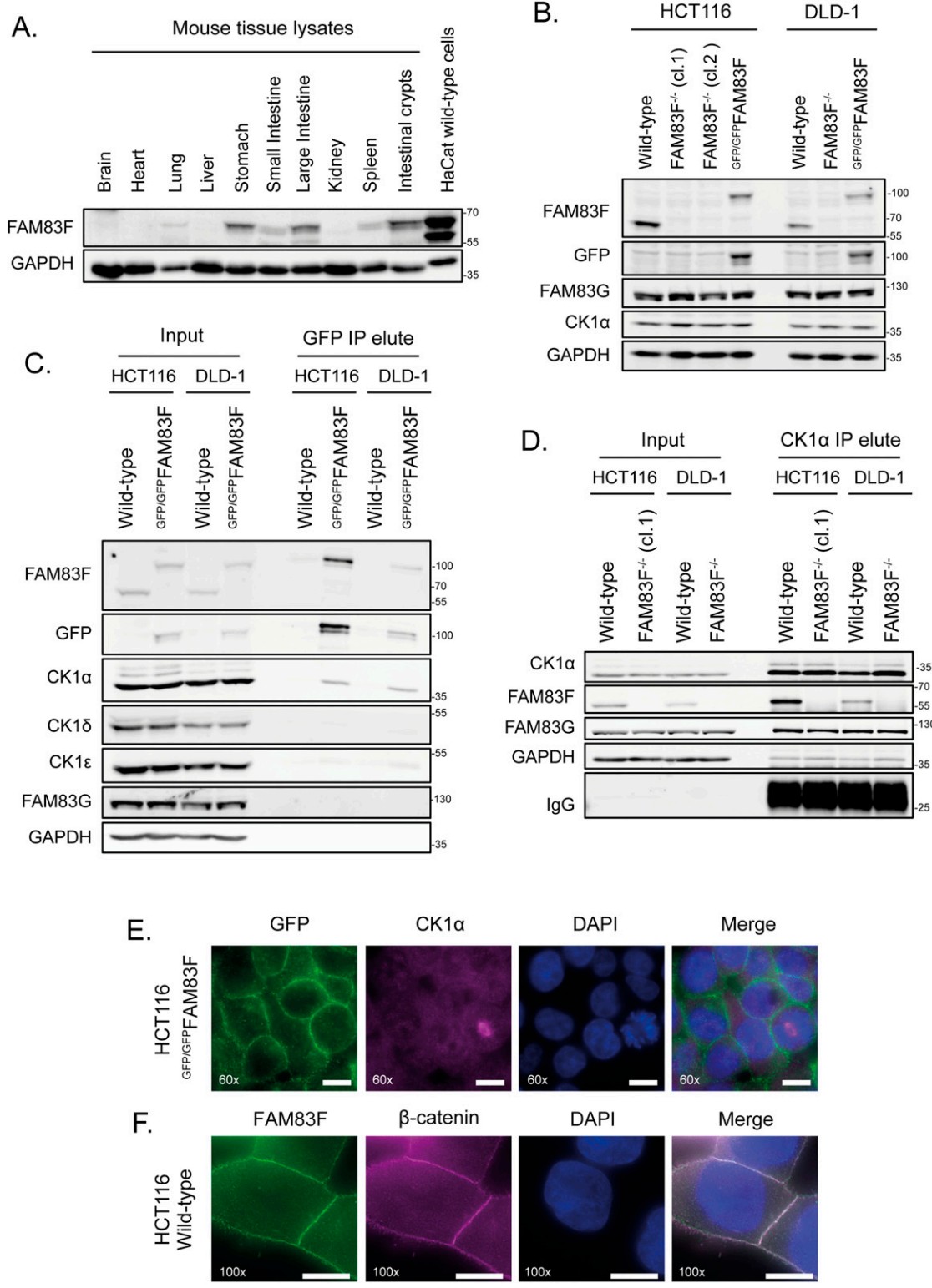

**Figure 3. Endogenous FAM83F localises to the plasma membrane and interacts with casein kinase 1α (CK1α).**
**(A)** Extractions from mouse tissues: brain, heart, lung, liver, stomach, small intestine, large intestine, kidney, spleen, and intestinal crypts, plus HaCaT wild-type cells were resolved by SDS–PAGE and subjected to Western blotting with indicated antibodies. **(B)** HCT116 wild-type, HCT116 FAM83F$^{-/-}$ (cl.1), HCT116 FAM83F$^{-/-}$ (cl.2), HCT116 $^{GFP/GFP}$FAM83F, DLD-1 wild-type, DLD-1 FAM83F$^{-/-}$, and DLD1 $^{GFP/GFP}$FAM83F cell extracts were resolved by SDS–PAGE and subjected to Western blotting with indicated antibodies. **(C)** HCT116 wild-type, HCT116 $^{GFP/GFP}$FAM83F, DLD-1 wild-type, and DLD-1 $^{GFP/GFP}$FAM83F cell extracts were subjected to immunoprecipitation with GFP trap beads. Input lysates and GFP IPs were resolved by SDS–PAGE and subjected to Western blotting with indicated antibodies. **(D)** HCT116 wild-type, HCT116 FAM83F$^{-/-}$ (cl.1),

cytoplasmic β-catenin (Fig S5) and as such an increase in Wnt signalling (Fig 2D). These conflicting results highlight the caveats associated with evaluating cell signalling in overexpression models. At an endogenous level, we show that absence of farnesylated FAM83F reduces membranous CK1α protein abundance and Wnt signalling output, indicating that membrane bound FAM83F is required for both membrane localisation of CK1α and mediating canonical Wnt signalling.

## Discussion

Little is known about the function of FAM83F and in most tissues and cell lines, except for the gastrointestinal tissues and colorectal cancer cell lines, FAM83F protein levels are undetectable. Interestingly, gastrointestinal tissues and colorectal cancer cells are reliant on canonical Wnt signalling for homeostasis and maintenance. In this study, we show that FAM83F mediates the canonical Wnt/β-catenin signalling pathway both in developing *Xenopus* embryos and in human cancer cells. We show that FAM83F is localised at the plasma membrane through farnesylation and directs CK1α to the plasma membrane. The FAM83F-CK1α complex appears to act upstream of GSK-3β, and we show that the interaction between FAM83F and CK1α and their membrane localisation is essential for driving Wnt signalling.

CK1 isoforms have been implicated in both positive and negative regulation of Wnt/β-catenin signalling (3). This suggests that the activity of CK1 isoforms is tightly regulated in a spatiotemporal manner. As key regulators of the CK1 isoforms, it is highly likely that the FAM83 proteins play a role. Any perturbation of the homeostatic balance of endogenous CK1 pools in cells, caused, for example, by overexpression of interacting proteins, is thus likely to disrupt coordinated roles of CK1 isoforms in Wnt signalling. This explains the apparent contradictory observations we made when overexpressing FAM83F^C497A in *Xenopus* embryos and in cells caused an increase in canonical Wnt signalling, whereas the endogenous knock-in of FAM83F^C497A mutant attenuated Wnt signalling. Expressing high levels of FAM83F^C497A protein, which redirects much of the endogenous CK1α protein to the nucleus, reduces the cytoplasmic pool of CK1α, thus removing inhibitory CK1α from the β-catenin destruction complex and thereby activating the canonical Wnt signalling pathway.

We have shown that FAM83F and FAM83G/PAWS1 both activate Wnt signalling, but FAM83F acts upstream of GSK-3β, whereas FAM83G/PAWS1 acts downstream (6). The specific CK1α substrates within the canonical Wnt signalling pathway which mediate the effects of both FAM83F and FAM83G/PAWS1 are unknown. Phosphorylation of β-catenin by CK1α at Ser45 is the most

established regulatory role of CK1α within the Wnt pathway and is critical for priming subsequent GSK-3 phosphorylation, ubiquitination, and degradation of β-catenin and thus inhibition of Wnt signalling (28). P120-catenin is one of the few reported Wnt-dependent CK1α substrates located at the plasma membrane. The sequential phosphorylation of P120-catenin, at the adherens junctions by CK1ε and CK1α is required for the internalisation of the Wnt signalosome (17). The contradictory nature of the roles of CK1 isoforms in Wnt signalling can be demonstrated by the CK1δ/ε isoforms which have been reported to phosphorylate multiple proteins within the Wnt signalling pathway including Dishevelled (29) and a co-transcriptional regulator, lymphoid enhanced binding factor 1 (Lef-1) (30). Interestingly, phosphorylation of Dishevelled at the membrane following Wnt stimulation promotes signalling, whereas phosphorylation of Lef-1 in the nucleus is inhibitory; thus, the same kinase can have opposing actions depending on subcellular localisation. It will be interesting to determine whether and how different FAM83 proteins coordinate the phosphorylation of these and other CK1 substrates to fine tune Wnt signalling.

FAM83F has been implicated in oncogenesis in several cancers (12, 14, 31), and increased abundance of FAM83F protein is also associated with a more aggressive phenotype and poor prognosis in oesophageal carcinoma (32). However, a mechanistic explanation for these oncogenic effects has not been forthcoming. We show here that overexpression of FAM83F protein in cells increases both basal and ligand-dependent canonical Wnt signalling which may explain the reported increase in cell proliferation. FAM83F has also been implicated in the stabilisation of p53 protein, a crucial tumour suppressor, by decreasing p53 ubiquitination and degradation (33). This stabilisation is also apparent with mutant p53 protein, thus FAM83F may have a tumour suppressor or an oncogenic role depending on the p53 mutational status of the cell (33). Interestingly, CK1α has also been reported to influence p53 stabilisation through binding to the E3 ligases MDM2 (34) and MDMX (35) which inhibit p53 activity through ubiquitination and direct binding, respectively. Therefore, these reported p53 effects may potentially be another function of the FAM83F-CK1α complex. We propose that FAM83F's biological role is to mediate the localisation of a proportion of CK1α and, thus, to facilitate phosphorylation of a subset of CK1α substrates. Therefore, FAM83F may have effects in multiple signalling pathways, beyond Wnt signalling and p53 activity, through determining CK1α localisation and its substrates. Although targeting CK1α therapeutically would have multiple unwanted consequences, the potential to inhibit specific pools of CK1α and potentially specific substrates through inhibition of the FAM83F-CK1α complex could make FAM83F an attractive therapeutic target in cancer.

DLD-1 wild-type, and DLD-1 FAM83F^−/− cell extracts were subjected to immunoprecipitation with anti-CK1α antibody. Input lysates and CK1α IPs were resolved by SDS–PAGE and subjected to Western blotting with indicated antibodies. **(E)** Representative wide-field immunofluorescence microscopy images of HCT116 ^GFP/GFP^FAM83F cells, labelled with antibodies recognising GFP (far left panels, green), CK1α (second row of panels from left, magenta), and DAPI (third row of panels from left, blue). Overlay of GFP, CK1α, and DAPI images as a merge is shown on the right. Immunofluorescence images captured with a 60× objective. **(F)** Representative wide-field immunofluorescence microscopy images of HCT116 wild-type cells, labelled with antibodies recognising FAM83F (far left panels, green), β-catenin (second row of panels from left, magenta), and DAPI (third row of panels from left, blue). Overlay of FAM83F, β-catenin, and DAPI images as a merge is shown on the right. Immunofluorescence images captured with a 100× objective. Scale bar represents 10 μm.
Source data are available for this figure.

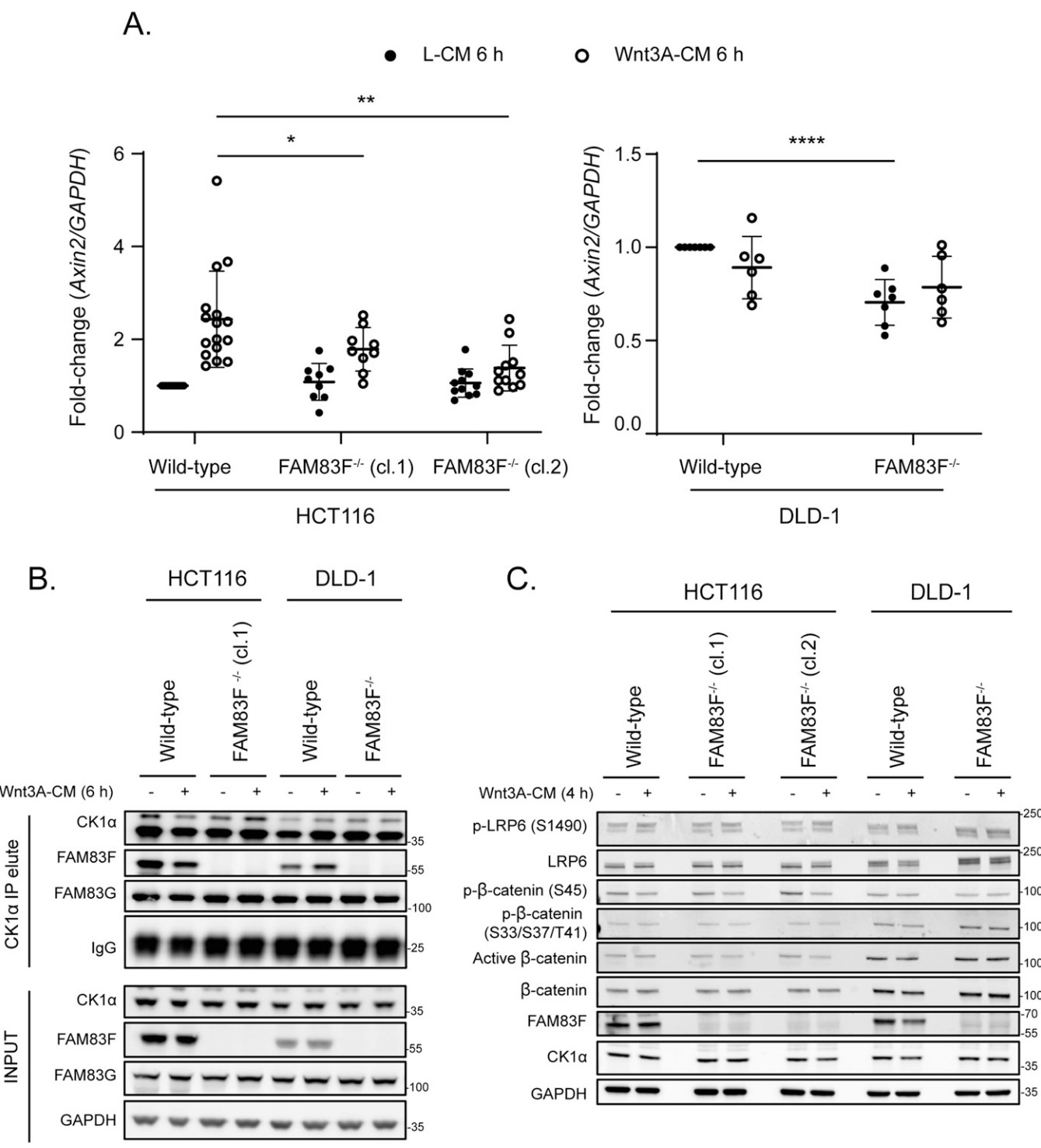

**Figure 4. Knockout of FAM83F reduces canonical Wnt signalling in colorectal cancer cells.**
**(A)** qRT-PCR was performed using cDNA from HCT116 wild-type, HCT116 FAM83F$^{-/-}$ (cl.1), HCT116 FAM83F$^{-/-}$ (cl.2), DLD-1 wild-type and DLD-1 FAM83F$^{-/-}$ cells following treatment with L-CM or Wnt3A-CM for 6 h, and primers for *Axin2* and *GAPDH* genes. *Axin2* mRNA expression was normalised to *GAPDH* mRNA expression and represented as fold change compared to L-CM treated wild-type cells. Data presented as scatter graph illustrating individual data points with an overlay of the mean ± SD. **(B)** HCT116 wild-type, HCT116 FAM83F$^{-/-}$ (cl.1), DLD-1 wild-type, and DLD-1 FAM83F$^{-/-}$ cells treated with L-CM or Wnt3A-CM for 6 h, were subjected to immunoprecipitation with casein kinase 1α (CK1α) antibody. Input lysates and CK1α IPs were resolved by SDS–PAGE and subjected to Western blotting with indicated antibodies. **(C)** Lysates from HCT116 wild-type, HCT116 FAM83F$^{-/-}$ (cl.1), HCT116 FAM83F$^{-/-}$ (cl.2), DLD-1 wild-type, and DLD-1 FAM83F$^{-/-}$ cells following treatment with L-CM or Wnt3A-CM for 4 h, were resolved by SDS–PAGE and subjected to Western blotting with indicated antibodies. Statistical analysis of (A) was completed using a Student's unpaired *t* test and comparing cell lines as denoted on graph. Statistically significant *P*-values are denoted by asterisks (**** < 0.0001, *** < 0.001, ** < 0.01, * < 0.05).
Source data are available for this figure.

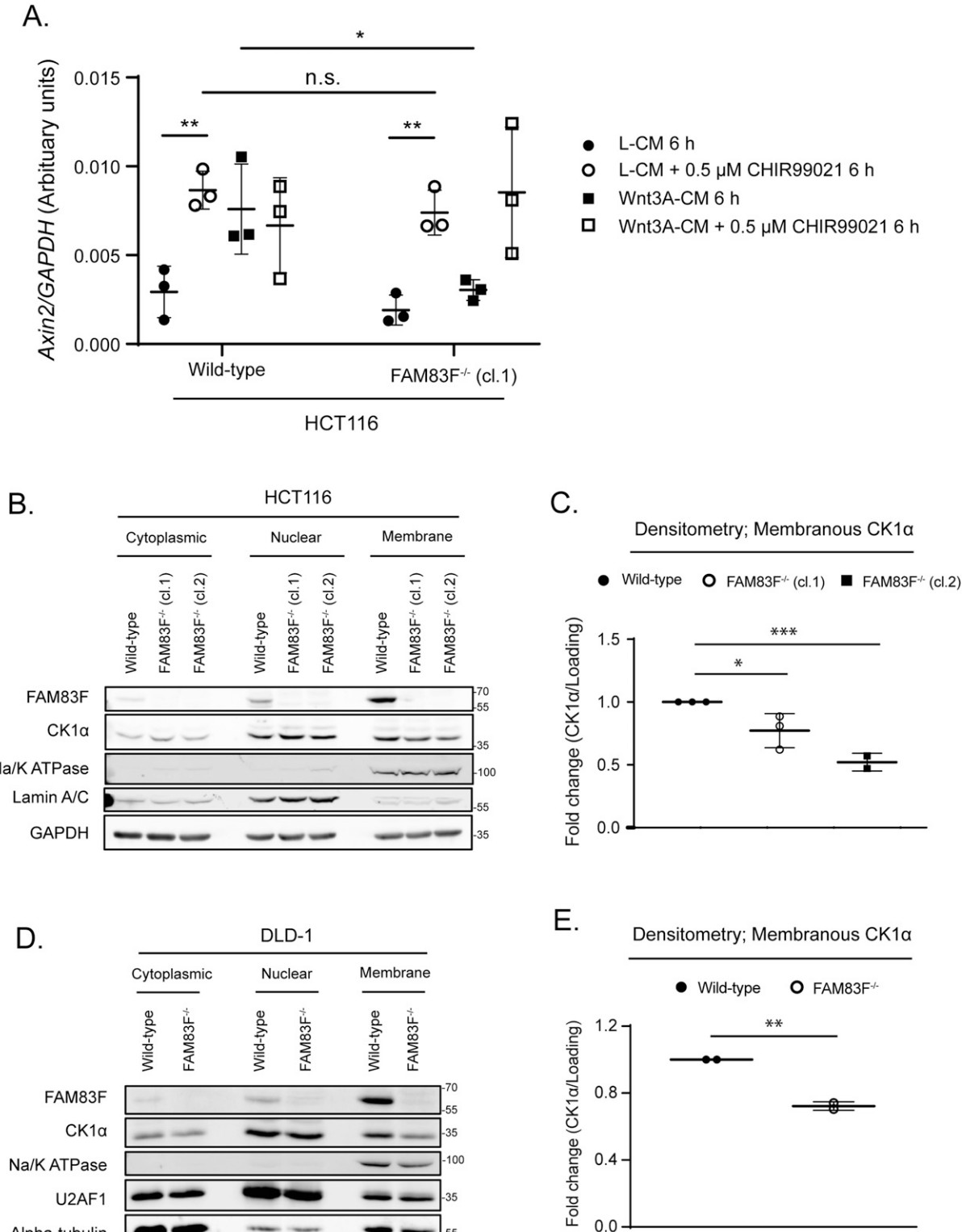

**Figure 5. FAM83F acts upstream of glycogen synthase kinase-3β and the loss of FAM83F protein reduces casein kinase 1α (CK1α) protein abundance at the plasma membrane.**

**(A)** qRT-PCR was performed using cDNA from HCT116 wild-type and HCT116 FAM83F$^{-/-}$ (cl.1) cell lines following treatment with L-CM or Wnt3A-CM with or without 0.5 μM CHIR99021 for 6 h, and primers for *Axin2* and *GAPDH* genes. *Axin2* mRNA expression was normalised to *GAPDH* mRNA expression and represented as arbitrary units. Data presented as scatter graph illustrating individual data points with an overlay of the mean ± SD. **(B)** Cytoplasmic, nuclear, and membrane lysates from HCT116 wild-type, HCT116 FAM83F$^{-/-}$ (cl.1), and HCT116 FAM83F$^{-/-}$ (cl.2) cell lines were resolved by SDS–PAGE and subjected to Western blotting with indicated antibodies. **(C)** Densitometry of

# Materials and Methods

## Plasmids

All constructs are available for request from the Medical Research Council-Protein Phosphorylation and Ubiquitylation Unit (MRC-PPU) reagents website (http://mrcppureagents.dundee.ac.uk) with the unique identifier (DU) numbers providing direct links to cloning strategy and sequence information. Sequences were verified by the DNA sequencing service, University of Dundee (http://www.dnaseq.co.uk). Constructs generated include; pcDNA5-FRT/TO GFP only (DU41455), pcDNA5-FRT/TO GFP FAM83A (DU44235), pcDNA5-FRT/TO GFP FAM83B (DU44236), pcDNA5-FRT/TO GFP FAM83C (DU42473), pcDNA5-FRT/TO GFP FAM83D (DU42446), pcDNA5-FRT/TO GFP FAM83E (DU44237), pcDNA5-FRT/TO GFP FAM83F (DU44238), pcDNA5-FRT/TO GFP FAM83G (DU33272), pcDNA5-FRT/TO GFP FAM83H (DU44239), pcDNA5-FRT/TO GFP FAM83F$^{C497A}$ (DU28157), pcDNA5-FRT/TO GFP FAM83F$^{D250A}$ (DU28268), pcDNA5-FRT/TO GFP FAM83F$^{F284A/F288A}$ (DU28260), pBABED.puro U6 FAM83F tv1 Nter KI sense (DU54050), pX335 FAM83F Nter KI Antisense (DU54056), pMSRQ FAM83F Nter GFP donor (DU54325), pBABED.puro U6 FAM83F ex2 KO sense (DU54848), pX335 FAM83F ex2 KO Antisense (DU54850), pBABED.puro U6 FAM83F Cter KI Sense A (DU60633), pX335 FAM83F Cter KI Antisense A (DU60635), and pMA FAM83F Cter C497A internal ribosome entry site (IRES) GFP donor (DU60711).

To amplify plasmids, 10 μl of *Escherichia coli* DH5α competent cells (Invitrogen) were transformed using 1 μl of plasmid DNA. The bacteria were incubated on ice for 10 min, heat-shocked at 42°C for 45 s, and then incubated in ice for a further 2 min. Transformed bacteria were spread on ampicillin (100 μg/ml) containing LB-agar medium plates and incubated at 37°C for 16 h. A 5-ml culture of ampicillin (100 μg/ml) containing LB medium was inoculated with a single bacteria colony and incubated at 37°C for 16 h with constant shaking. Following bacterial growth, the bacteria were pelleted and plasmid DNA was purified using QIAprep Spin Miniprep Kit (27104; Quigen) following the manufacturer's instructions. A NanoDrop 1000 spectrophotometer (Thermo Fisher Scientific) was used to determine the isolated DNA yield.

## Antibodies

Antibodies recognising GFP (S268B), CK1α (SA527), CK1δ (SA609), CK1ε (SA610), and FAM83F (SA103) are available on request from the MRC-PPU reagents website (http://mrcppureagents.dundee.ac.uk). Antibodies for β-actin (#4967), GAPDH (14C10) (#2118), p-LRP6 (S1490) (#2568), LRP6 (C47E12) (#3395), β-catenin (D10A8) (#8480), phospho-β-catenin (S45) (#9564), phospho-β-catenin (S33/S37/

T41) (#9561), Na/K ATPase (D4Y7E) (#23565), and Lamin A/C (#2032) were purchased from Cell Signalling Technology. Additional antibodies used were FAM83G (ab121750; Abcam), α-tubulin (MA1-80189; Thermo Fisher Scientific), Active-β-catenin (anti-ABC) clone 8E7 (05-665; END Millipore), mouse anti-GFP clone 7.1 and 13.1 (Roche), mouse-monoclonal anti-HA (H9658; Sigma-Aldrich), and U2AF1 (PA5-28510; Thermo Fisher Scientific). Secondary antibodies used were StarBright Blue 700 Goat anti-Rabbit IgG (12004161; Bio-Rad), StarBright Blue 700 Goat anti-Mouse IgG (12004158; Bio-Rad), IRDye 800CW Donkey anti-Goat IgG (926-32214; LI-COR), IRDye 800CW Goat anti-Rat IgG (926-32219; LI-COR), IRDye 800CW Goat anti-Mouse (926-32210; LI-COR), and IRDye 680LT Goat anti-Rabbit (926-68021; LI-COR).

## Constructs and mRNA synthesis for *Xenopus* assays

Coding sequences for the ORF of *fam83fa* were obtained from the Ensembl genome browser (Ensembl release 87—December 2016 EMBL-EBI, GRCz9). Primers were designed to amplify the ORFs to clone into pENTR/D-TOPO entry vectors (Invitrogen) according to the manufacturer's instructions. For *fam83fa*, the template was amplified from a previously prepared zebrafish cDNA library. Inserts were then subcloned into pCS2+ N′ HA tagged vectors that had been converted into Gateway (Invitrogen) destination vectors, according to the manufacturer's instructions. Mutant constructs (*fam83fa$^{F275A}$*, *fam83fa$^{F279A}$*, and *fam83fa$^{F275/279A}$*) were generated by site-directed mutagenesis using the Q5 Site-Directed Mutagenesis Kit (NEB) according to the manufacturer's instructions. C′ terminal truncation constructs (*fam83fa$^{1-500aa}$*, *fam83fa$^{1-400aa}$*, *fam83fa$^{1-356aa}$*, and *fam83fa$^{1-300aa\ (DUF)}$*) were generated by pENTR/D-TOPO cloning as previous, using reverse primers designed accordingly. All constructs were Sanger-sequenced by the Genomics Equipment Park Science Technology Platform at the Crick. For mRNA synthesis, PCR templates for in vitro transcription were generated from the pCS2+ N′ HA–tagged destination vectors. Templates were then used in an SP6 mMessage mMachine (Invitrogen) transcription reaction to generate capped mRNAs.

## *Xenopus laevis* maintenance, microinjection and Western blotting

All *X. laevis* work, including housing and husbandry, was undertaken in accordance with The Crick Use of Animals in Research Policy, the Animals (Scientific Procedures) Act 1986 (ASPA) implemented by the Home Office in the UK and the Animal Welfare Act 2006. Consideration was given to the "3Rs" in experimental design. *X. laevis* embryos were obtained by in vitro fertilisation and staged according to Nieuwkoop and Faber (1994) (36). Embryos were maintained in Normal Amphibian Medium (37) until the four-cell stage

CK1α protein abundance from (B) membrane lysates normalised to GAPDH protein abundance and represented as fold change compared with HCT116 wild-type cells. Data presented as scatter graph illustrating individual data points with an overlay of the mean ± SD. **(D)** Cytoplasmic, nuclear, and membrane lysates from DLD-1 wild-type and DLD-1 FAM83F$^{-/-}$ cell lines were resolved by SDS–PAGE and subjected to Western blotting with indicated antibodies. **(E)** Densitometry of CK1α protein abundance from (D) membrane lysates normalised to Na/K ATPase protein abundance and represented as fold change compared with DLD-1 wild-type cells. Data presented as scatter graph illustrating individual data points with an overlay of the mean ± SD. **(B, D)** The specificity of cytoplasmic, nuclear and membrane compartment lysates were determined by Western blotting with the following subcellular compartment-specific antibodies: α-tubulin (cytoplasmic), Lamin A/C (nuclear), U2AF1 (nuclear), and Na/K ATPase (membrane). **(A, C, E)** Statistical analysis was completed using a Student's unpaired *t* test and comparing cell lines as denoted on graphs. Statistically significant *P*-values are denoted by asterisks (**** < 0.0001, *** < 0.001, ** < 0.01, * < 0.05). Source data are available for this figure.

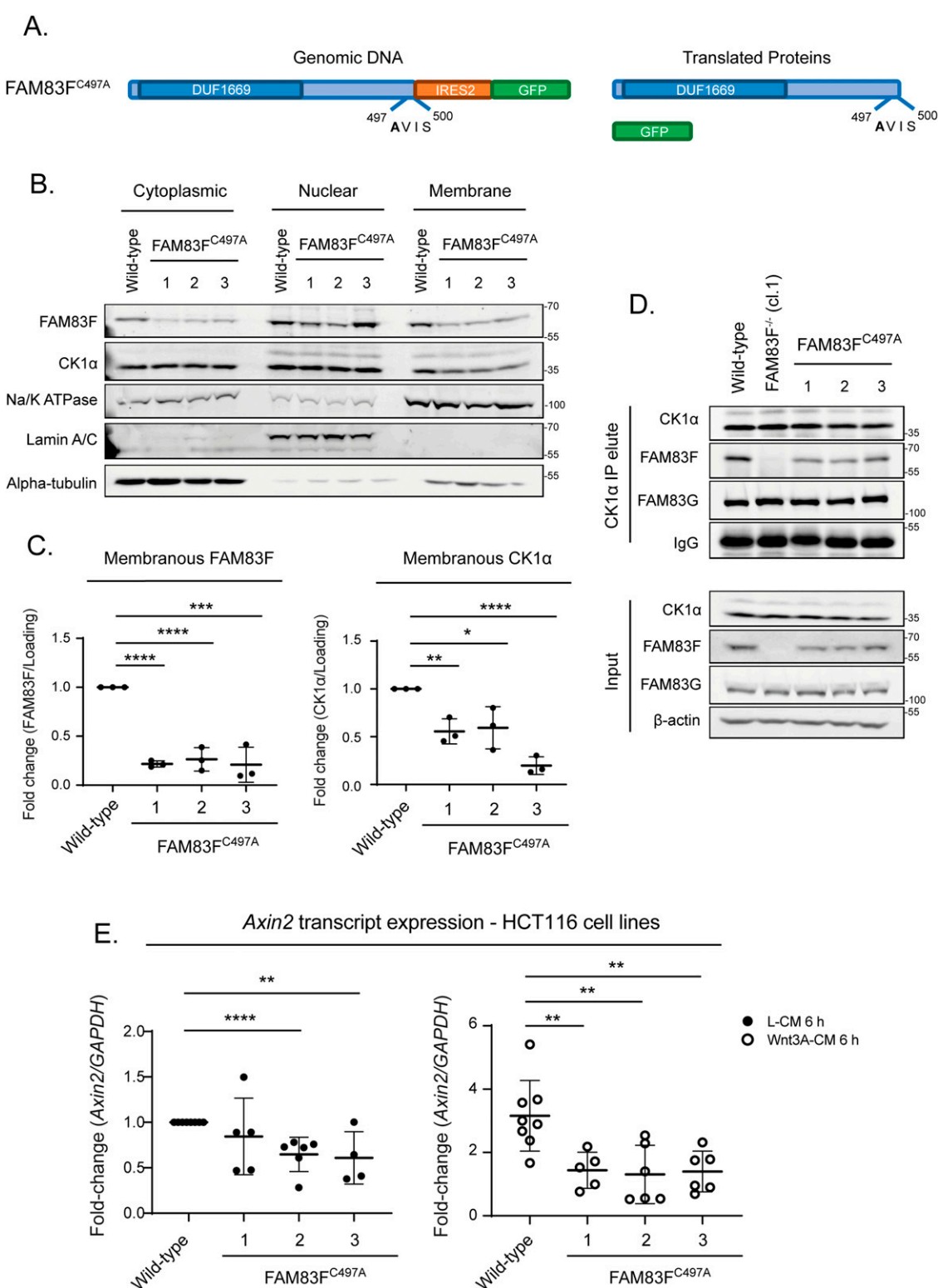

**Figure 6. Membranous localisation of FAM83F is required for FAM83F's role in canonical Wnt signalling.**
**(A)** Cartoon of the FAM83F$^{C497A}$ knock-in strategy illustrating the genomic DNA sequence in which the C497A point mutation is knocked-in to FAM83F along with an IRES2 (Internal ribosome entry sequence) and GFP coding sequence which are inserted after the FAM83F protein coding sequence and the eventual translated proteins.
**(B)** Cytoplasmic, nuclear and membrane lysates from HCT116 wild-type and HCT116 FAM83F$^{C497A}$ (clones 1–3) cell lines were resolved by SDS–PAGE and subjected to Western blotting with indicated antibodies. The specificity of cytoplasmic, nuclear and membrane compartment lysates was determined by Western blotting with the following subcellular compartment-specific antibodies: α-tubulin (cytoplasmic), Lamin A/C (nuclear), and Na/K ATPase (membrane). **(C)** Quantification of FAM83F and casein

was reached, at which point they were injected with 500 pg of the indicated capped mRNA into a single ventral blastomere. Embryos were allowed to develop until approximately stage 35 at which point they were fixed in 4% paraformaldehyde. Embryos were then counted and scored for secondary axial phenotype classes as shown.

For Western blotting, embryos were obtained as above and injected into the animal hemisphere at the one-cell stage with 500 pg of the indicated capped mRNA. Embryos were allowed to develop to stage 10 before being lysed in 10 μl/embryo ice-cold lysis buffer (1% IGEPAL, 150 mM NaCl, 10 mM Hepes, pH 7.4, 2 mM EDTA, and protease inhibitor cocktail [A32965; Pierce]). Lipids and yolk were removed from the lysate by FREON extraction (equal volume), and the aqueous phase was collected following centrifugation for 15 min at 4°C. Protein extracts were then denatured in SDS buffer before being separated by SDS–PAGE (or snap-frozen on dry ice and stored at –80°C for later). Proteins were then transferred onto a low-fluorescence polyvinylidene fluoride (LF-PVDF) (LI-COR) membrane and blocked in Odyssey blocking buffer in TBS (LI-COR) for 1 h at RT. Blots were then incubated overnight at 4°C in primary antibodies diluted in Odyssey blocking buffer. Following TBST washes, blots were then incubated in secondary antibodies diluted in TBST with 0.02% SDS. Blots were then imaged using the LI-COR Odyssey imaging system in 700 and 800 nm channels and processed for brightness and contrast in Adobe Photoshop CC (2019).

## Cell culture

U2OS (HTB-96; ATCC), HCT116 (CCL-247; ATCC), DLD-1 (CCL-221; ATCC), mouse fibroblast L-cells that stably overexpress Wnt3A (CRL-2647; ATCC), mouse fibroblast L cells (CRL-2648; ATCC), Flp-In T-Rex U2OS (which were created using Flp-In T-Rex Core kit [K650001; Thermo Fisher Scientific] and have been previously reported (1)), Flp-In T-Rex HEK293 (R78007; Thermo Fisher Scientific) and HaCaT (obtained from Joan Massague's lab at Memorial Sloan Kettering Cancer Centre, not commercially available but can be provided on request) (38) cells were maintained in DMEM (Gibco) containing 10% FCS (Hyclone), 100 U/ml penicillin (Lonza), 100 mg/ml streptomycin (Lonza), and 2 mM L-glutamine (Lonza). Cell lines were routinely tested for mycoplasma contamination and only mycoplasma-free cell lines were used for experimentation. The additional cell lines reported in Fig S6 were kindly donated by various research groups within the MRC-PPU facility (University of Dundee) in the form of cell pellets and were immediately lysed to determine protein abundance.

## Generation of stable Flp-In T-Rex cell lines

The Flp-In T-Rex U2OS were transfected with the N-terminal GFP-tagged FAM83A, FAM83B, FAM83C, FAM83D, FAM83E, FAM83F, FAM83G,

FAM83H, FAM83F$^{C497A}$, FAM83F$^{D250A}$, FAM83F$^{F284A/F288A}$, or GFP only packaged in a pcDNA5-FRT/TO vector, along with Flp recombinase pOG44 (Invitrogen) in a ratio of 1:9 μg. The Flp-In T-Rex HEK293 cells were transfected with the N-terminal GFP-tagged FAM83F packaged in a pcDNA5-FRT/TO vector along with Flp recombinase pOG44 (Invitrogen) in a ratio of 1:9 μg. Plasmids were diluted in 1 ml of OptiMem (Gibco) and 20 μl polyethylenimine (PEI; 1 mg/ml) (Polysciences) was added. The transfection mixture was vortexed and incubated for 20 min at room temperature before adding dropwise to a 10-cm diameter dish of cells in complete culture medium. Selection of cells was performed 24 h post transfection with the addition of 50 μg/ml hygromycin and 15 μg/ml blasticidin to complete culture medium. Resistant cells were grown to confluency and tested for expression by Western blotting. Expression was induced by incubating cells in 20 ng/ml doxycycline for 24 h.

## Generation of FAM83F$^{−/−}$, $^{GFP/GFP}$FAM83F and FAM83F$^{C497A}$ cell lines using CRISPR/Cas9

All CRISPR/Cas9 technology procedures were performed using a dual guide nickase approach. FAM83F knock-out HCT116 (clone.1), DLD-1, U2OS and HaCaT cell lines were generated by targeting the *FAM83F* locus with sense guide RNA (pBabeD-puro vector, DU54848); GCGTCCAGGATGATGTACACT and antisense guide RNA (pX335-Cas9-D10A vector, DU54850); GGCAGGAGTGAAGTATTTCC. N-terminal GFP knock-in to the *FAM83F* locus in HCT116 and DLD-1 cells were generated by targeting the *FAM83F* locus with sense guide RNA (pBabeD-puro vector, DU54050); GTTCAGCTGGGACTCGGCCA, antisense guide RNA (pX335-Cas9-D10A vector, DU54056); GCGAGGCG-CACGTGAACGAGA and the GFP-FAM83F donor (pMK-RQ vector, DU54325). FAM83F$^{−/−}$ HCT116 (clone.2) was generated by the targeting of the N terminus of FAM83F with the intention of knocking in a GFP tag, but this clone did not incorporate the GFP donor plasmid, and after inefficient repair of the cut DNA, the clone was null for FAM83F protein expression. HCT116 FAM83F$^{C497A}$ knock-in cell lines were generated by targeting the C-terminal of *FAM83F* with the sense guide RNA (pBabeD-puro vector, DU60633), antisense guide RNA (pX335-Cas9-D10A vector, DU60635), and FAM83F C-terminal donor containing C497A mutation (pMA vector, DU60711). The donor plasmid contained an IRES and GFP sequence after the FAM83F stop codon to aid selection of positive clones.

For transfection, plasmids (1 μg of guide RNAs ±3 μg donor) were diluted in 1 ml OptiMem (Gibco) and 20 μl PEI (1 mg/ml). The transfection mixture was vortexed for 15 s and incubated for 20 min at room temperature. This mixture was then added dropwise to a 10-cm diameter dish containing ~70% confluent cells in complete culture medium. Transfected cells were selected 24 h post-transfection

---

kinase 1α (CK1α) protein abundance in membrane enriched fractions from (B). FAM83F and CK1α protein abundance is normalised to loading control and presented as fold-change compared with HCT116 wild-type cells. Data presented as scatter graph illustrating individual data points with an overlay of the mean ± SD. **(D)** Cell lysates from HCT116 wild-type, HCT116 FAM83F$^{−/−}$ (cl.1), and HCT116 FAM83F$^{C497A}$ (clones 1–3) cell lines were subjected to immunoprecipitation with anti-CK1α antibody. Input lysates and CK1α IP elutes were resolved by SDS–PAGE and subjected to Western blotting with indicated antibodies. **(E)** qRT-PCR was performed using cDNA from HCT116 wild-type and HCT116 FAM83F$^{C497A}$ (clones 1–4) cell lines following treatment with L-CM or Wnt3A-CM for 6 h, and primers for *Axin2* and *GAPDH* genes. *Axin2* mRNA expression was normalised to *GAPDH* mRNA expression and represented as fold change compared with L-CM–treated wild-type cells. Data presented as scatter graph illustrating individual data points with an overlay of the mean ± SD. **(C, E)** Statistical analysis was completed using a Student's unpaired *t* test and comparing cell lines as denoted on graphs. Statistically significant *P*-values are denoted by asterisks (**** < 0.0001, *** < 0.001, ** < 0.01, * < 0.05). Source data are available for this figure.

with the addition of 2 μg/ml puromycin to complete culture medium for 48 h. Single cells were isolated by FACS with single GFP-positive cells (for knock-in strategies) or all single cells (for knock-out strategies) plated in individual wells of a 96-well plate, pre-coated with 1% (wt/vol) gelatin (Sigma-Aldrich). Viable clones were expanded and screened by Western blotting for efficient knock-in or knock-out.

Knock-in and knock-out clones were verified by DNA sequencing. DNA was isolated from cell pellets using the DNeasy Blood & Tissue kit (69505; QIAGEN). Primers were generated to amplify the surrounding region of the guide RNA target sites with the following primer pairs: FAM83F KO exon 2 (forward: TCATTGCTGTGGTCATGGAC, reverse: AATCCGGAAGTCAGTGAGCT), FAM83F N-terminal GFP KI (forward: TGCGCGGAAAATGAACTCGTACC, reverse: GTAGAAACCAGTGTCCGTCCAGC), and FAM83F C-terminal C497A KI (forward: GCTGAATCCACCAAGCGTT, reverse: CGTGTGACTGAGATGCTTCG). The region was amplified by PCR with KOD Hot Start Polymerase (Merck) according to the manufacturer's instructions. The PCR products were visualised on a 1.5% agarose gel using SYBR Safe DNA gel stain (Invitrogen) and 100-bp and 1-kbp DNA ladders (Promega). The PCR products of positive clones were then cloned into competent cells using the StrataClone PCR Cloning Kit (Agilent) according to the manufacturer's protocol. Isolation of DNA and sequencing was performed by the MRC-PPU DNA sequencing and services (http://mrcppureagents.dundee.ac.uk).

### Generation of L- and Wnt3A-conditioned medium

Conditioned media were generated from mouse fibroblast L-cells and mouse fibroblast L-cells that stably overexpress Wnt3A. L-cells and L-Wnt3A cells were grown in DMEM in 15-cm diameter dishes for 3 d before medium was filtered (0.22 μm) and stored as L-conditioned medium (L-CM) and Wnt3A-conditioned medium (Wnt3A-CM). Conditioned medium was diluted 50:50 in DMEM containing 10% FCS before use.

### Compound treatments

CHIR99021 (Tocris), a highly selective GSK-3 inhibitor, was added to cells at a concentration of 0.5 μM for 6 h before lysis.

### Protein extraction from cells

Cells were washed in PBS twice, scraped in PBS, and pelleted. For whole-cell protein extractions, cell pellets were resuspended in total lysis buffer (20 mM Tris–HCl [pH 7.5], 150 mM NaCl, 1 mM $Na_2EDTA$, 1 mM EGTA, 1% [vol/vol] Triton X-100, 2.5 mM sodium pyrophosphate, 1 mM $\beta$-glycerophosphate, 1 mM $Na_3VO_4$, and 1× complete EDTA-free protease inhibitor cocktail [Roche]). Lysates were incubated on ice for 30 min and vortexed regularly then clarified at 17,000$g$ for 20 min. For cellular fractionation, cell pellets were washed in PBS twice, scraped in PBS, pelleted, and then separated into cytoplasmic, nuclear, and membrane lysates using a subcellular protein fractionation kit (Thermo Fisher Scientific) following the manufacturer's protocol. Briefly, cell pellets were resuspended in sequential buffers and clarified to isolate specific cellular compartments (cytoplasmic, nuclear, membrane, and cytoskeletal).

For cytoplasmic/nuclear protein extractions, cell pellets resuspended in cytoplasmic lysis buffer (20 mM Tris–HCL [pH 7.5], 0.1 mM EDTA, 2 mM $MgCl_2$, 1% NP40, 50 nM $\beta$-glycerophosphate, and 1× complete EDTA-free protease inhibitor cocktail [Roche]) and incubated for 2 min at room temperature, then 10 min on ice, before clarifying lysate at 1,000$g$ for 5 min. The supernatant was collected as the cytoplasmic extract before the residual pellet was washed in wash buffer (20 mM Tris–HCL [pH 7.5], 0.1 mM EDTA, 2 mM $MgCl_2$, 50 nM $\beta$-glycerophosphate, and 1× complete EDTA-free protease inhibitor cocktail [Roche]) three times. The residual pellet was resuspended in nuclear lysis buffer (20 mM Hepes, 0.4 M NaCl, 25% glycerol, 1 mM EDTA, 0.5 mM NaF, 0.5 mM $Na_3VO_4$, 0.5 mM DTT, and 1× complete EDTA-free protease inhibitor cocktail [Roche]) and the pellet disrupted by three quick freeze/thaw cycles then incubated on ice for 30 min. Lysate was clarified at 17,000$g$ for 20 min at 4°C with the supernatant collected as the nuclear extract.

### Protein extraction from mouse tissue

Mouse tissue samples were obtained from a single male C57BL/6j mouse, which was obtained from the MRC-PPU after it was designated as surplus to current breeding requirements and culled by schedule one methods. Tissue samples were dissected, washed in PBS and snap frozen in liquid nitrogen. Frozen tissue samples were ground using a mortar and pestle until the sample was a fine powder which was then resuspended in PBS and pelleted. This pellet was resuspended in total lysis buffer (20 mM Tris–HCl [pH 7.5], 150 mM NaCl, 1 mM $Na_2EDTA$, 1 mM EGTA, 1% [vol/vol] Triton X-100, 2.5 mM sodium pyrophosphate, 1 mM $\beta$-glycerophosphate, 1 mM $Na_3VO_4$, and 1× complete EDTA-free protease inhibitor cocktail [Roche]). Lysates were incubated on ice for 45 min and vortexed regularly before clarifying at 17,000$g$ for 20 min at 4°C.

For the isolation of single crypts from the mouse small intestine, a section of proximal small intestine was washed in PBS then cut into small fragments. Further washing of the fragments in PBS was completed until minimal contamination remained. The intestinal fragments were incubated in 3 mM EDTA for 30 min at 4°C on a rotating wheel to dissociate the crypts. The EDTA was removed and the fragments gently washed in PBS before washing the fragments more aggressively in PBS to dislodge the crypts. Crypts were collected as a component of the supernatant. Crypts were pelleted at 100$g$ for 5 min. The pellet was resuspended in total cell lysis buffer (20 mM Tris–HCl [pH 7.5], 150 mM NaCl, 1 mM $Na_2EDTA$, 1 mM EGTA, 1% [vol/vol] Triton X-100, 2.5 mM sodium pyrophosphate, 1 mM $\beta$-glycerophosphate, 1 mM $Na_3VO_4$, and 1× complete EDTA-free protease inhibitor cocktail [Roche]). Lysates were incubated on ice for 30 min with regular vortexing and clarified at 17,000$g$ for 20 min at 4°C.

### SDS–PAGE and Western blotting

The protein concentration of lysates was measured using the Pierce Coomassie Bradford Protein Assay Kit (Thermo Fisher Scientific). Final protein concentrations were adjusted to 1–3 μg/μl in lysis buffer and NuPAGE 4× lithium dodecyl sulphate sample buffer (NP0007; Thermo Fisher Scientific) was added to a final concentration of 1× and lysates were denatured at 95°C for 5 min. Lysates

(20–40 μg protein) were separated by SDS–PAGE gels and transferred to 0.2-μm pore size nitrocellulose membrane (1620112; Bio-Rad). Following washing in TBS-T (50 mM Tris–HCL [pH 7.5], 150 mM NaCl, and 0.1% [vol/vol] Tween 20) membranes were incubated in 5% (wt/vol) milk in PBS for 60 min. Membranes were washed in TBS-T, then incubated in primary antibody (diluted 1:500–1:1,000 in 5% [vol/vol] milk in TBS-T) for 16 h at 4°C. Membranes were washed in TBS-T (3 × 10 min), incubated in secondary antibody (diluted 1:5,000 in 5% [vol/vol] milk in TBS-T) for 1 h at room temperature and then washed in TBS-T (3 × 10 min). Detection of fluorescent secondary antibody was performed using the Chemidoc system (Bio-Rad) and Image Lab software (Bio-Rad). Densitometry of protein blots was completed, using ImageJ software (https://imagej.net), by measuring the density of protein of interest bands and normalising to that of the corresponding loading control bands. For subcellular fractions, 15 μg of protein was loaded for each fraction and the corresponding compartment-specific loading control was used for densitometry normalisation. Statistical analysis and preparation of graphs was completed using Microsoft Excel software (www.microsoft.com) and Prism 8 (www.graphpad.com), respectively.

## Immunoprecipitation

Lysates were prepared and protein concentrations quantified as described previously. For GFP immunoprecipitations (IPs), 10 μl of pre-equilibrated GFP-Trap Agarose beads (ChromoTek) were added to each lysate sample (1 mg protein) and incubated on a rotating wheel for 16 h at 4°C. For antibody-IPs, anti-CK1α (1 μg) antibody was added to each lysate sample (1 mg protein), incubated on a rotating wheel for 16 h at 4°C, and then 20 μl of pre-equilibrated Protein G Sepharose beads (50% beads: lysis buffer slurry) (DSTT) were added to each lysate sample and incubated on a rotating wheel for 1 h at 4°C. Following incubation of the lysate with beads, the beads were pelleted, and the supernatant was removed and stored as flow-through. Beads were washed in lysis buffer three times before eluting proteins by the addition of 20–40 μl of NuPAGE 1× lithium dodecyl sulphate sample buffer to the beads and denaturing proteins by incubating at 95°C for 10 min. The input (IN) and eluted samples (IP) were separated by SDS–PAGE and subjected to Western blotting with antibodies as previously outlined.

## Immunofluorescence

Cells were plated in 12-well plates containing sterile glass coverslips. Cells were incubated for 24 h before removing medium and incubating cells in 4% paraformaldehyde for 15 min to fix. Following fixation, cells were permeabilized in 0.2% Triton X-100 for 10 min, incubated in blocking buffer (5% [wt/vol] bovine serum albumin in PBS) for 60 min and then washed in TBS-T before incubating in primary antibody. The following primary antibodies were diluted in 0.5% (wt/vol) bovine serum albumin in TBS-T and incubated for 16 h at 4°C: mouse anti-GFP (1:1,000) (Roche), sheep anti-CK1α (1:100) (DSTT), rabbit anti–β-catenin (1:250) (CST) or sheep anti-FAM83F (1:250) (DSTT). Coverslips were washed in PBS/0.1% Tween-20 (3 × 10 min) then incubated in Alexa Fluor 488 phalloidin, Alexa Fluor donkey anti-mouse 488 secondary, Alexa Fluor donkey anti-sheep 594 secondary, Alexa Fluor donkey anti-sheep 488 secondary or

Alexa Fluor donkey anti-rabbit 594 secondary (Thermo Fisher Scientific) diluted (1:500) in 0.5% (wt/vol) bovine serum albumin in PBS/0.1% Tween-20 for 60 min at room temperature. Coverslips were washed in PBS/0.1% Tween-20 (3 × 10 min), incubated in 1 μg/ml DAPI diluted in PBS/0.1% Tween-20 for 5 min at room temperature before further washing in PBS/0.1% Tween-20 (3 × 5 min). Coverslips were mounted and sealed on glass slides using Vectashield mounting medium (Vector Laboratories) and CoverGrip coverslip sealant (Biotium), respectively. Fluorescence images were captured using a Deltavision microscope with a 20×, 60× or 100× objective. Images were prepared for publication using the Omero software (www.openmicroscopy.org).

## Dual luciferase reporter assays

Cells were plated in six-well plates and grown to ~70% confluence in complete culture medium. Cells were transfected with either 500 ng of Super TOPFlash (Addgene) or Super FOPFlash (Addgene) luciferase plasmids plus 10 ng of Renilla (Addgene) luciferase plasmids. Plasmids were diluted in 1 ml OptiMem (Gibco) and 20 μl of PEI (1 mg/ml) was added. The transfection mixture was vortexed then incubated for 20 min at room temperature before adding dropwise to a six-well plate of cells in complete culture medium. Following transfection (24 h), cells were treated with either L-conditioned medium or L-Wnt3A-conditioned medium for 6 h. Cells were washed in PBS twice and lysed in passive lysis buffer (#E194A; Promega) for 15 min on a rocker. After lysis, 20 μl of lysate was transferred to triplicate wells in a white bottom 96-well plate and then 20 μl of 2× luciferase assay buffer (50 mM Tris/Phosphate, 16 mM MgCl$_2$, 2 mM DTT, 1 mM ATP, 30% [vol/vol] Glycerol, 1% [wt/vol] BSA, 0.25 mM D-luciferin, 8 μM Sodium Pyrophosphate) was added to each well, incubated for 2 min, and absorbance at 560 nm measured. Immediately after reading absorbance, 20 μl of 3× Renilla buffer (45 mM Na$_2$EDTA, 30 mM Sodium Pyrophosphate, 1.425 M NaCl, 0.06 mM PTC124, 0.01 mM h-CTZ) was added to each well and incubated for 5 min before absorbance at 560 nm was again measured. The luciferase absorbance counts were normalised to Renilla absorbance counts, which represent a measure of transfection efficiency.

## Quantitative real-time PCR (qRT-PCR)

Cells were plated in six-well plates and grown to ~70% confluence in complete culture medium, then conditioned-medium or treatment added as indicated. RNA extractions were completed using RNeasy mini kit (74104; QIAGEN) following the manufacturer's instructions and RNA was quantified using a NanoDrop 3300 Fluorospectrometer (Thermo Fisher Scientific). Synthesis of cDNA was completed using 1 μg of RNA and the iScript cDNA synthesis kit (Bio-Rad). Each qRT-PCR was performed as triplicate reactions with the following reaction mixture: 2 μM forward primer, 2 μM reverse primer, 50% (vol/vol) iQ SYBR green supermix (Bio-Rad), and 2 μl cDNA (diluted 1:5) in a 10 μl final volume. Reactions were completed on a CFX384 real-time system qRT-PCR machine (Bio-Rad). Primers were designed using Benchling (www.benchling.com) and purchased from Invitrogen. *Axin2* forward: TACACTCCTTATTGGGCGATCA, *Axin2* reverse: TTGGCTACTCGTAAAGTTTTGGT, *GAPDH* forward: TGCAC-CACCAACTGCTTAGC, *GAPDH* reverse: GGCATGGACTGTGGTCATGAG. The comparative Ct method (ΔΔCt Method) was used to analyse the

datasets with *Axin2* the Wnt target gene and *GAPDH* the endogenous control gene. Statistical analysis and preparation of graphs was completed using Microsoft Excel software (www.microsoft.com) and Prism 8 (www.graphpad.com), respectively.

## Mass spectrometry

Mass spectrometry was performed as previously described (1). Briefly, expression of GFP-FAM83F in HEK-293 Flp/Trx cells was induced with doxycycline 24 h before lysis. Cells were lysed and proteins were immunoprecipitated using GFP-Trap Agarose beads (ChromoTek) as previously outlined. Proteins were separated by 4–12% gradient SDS–PAGE, stained with InstantBlue and gel slices covering each lane were excised and trypsin-digested. Mass spectrometry analysis of the peptides was performed by LC-MS/MS on the Linear Ion Trap-Orbitrap Hybris Mass Spectrometer (Orbitrap Velos Pro; Thermo Fisher Scientific) coupled to a U3000 RSLC Hplc (Thermo Fisher Scientific). Peptides were trapped on a nanoViper Trap column and (2 cm × 100 $\mu m$, C18, 5 $\mu m$, 100 Å; Thermo Fisher Scientific) and then separated on a 15 cm EASY-spray column (ES800; Thermo Fisher Scientific) equilibrated with a flow of 300 nl/ml of 3% solvent B (80% acetonitrile, 0.08% formic acis, and 3% DMSO in $H_2O$). The following elution gradient was completed: time (min)/solvent B (%) (0:3, 5:3, 45:35, 47:99, 52:99, 55:3, 60:3). Data were acquired in the data-dependent mode, automatically switching between MS and MS/MS acquisition. Full-scan spectra (mass/charge ratio [$m/z$] = 400–1,600) were acquired in the Orbitrap with resolution $R$ = 60,000 at $m/z$ 400 (after accumulation to a Fourier Transform Mass Spectrometry Full Automatic Gain Control [AGC] Target value of 1,000,000 and an Fourier Transform Mass Spectrometry Msn AGC Target value of 50,000). The 20 most intense ions, above a specified minimum signal threshold (2,000), based on a low-resolution (R = 15,000) preview of the survey scan, were fragmented by collision-induced dissociation and recorded in the linear ion trap (Full AGC Target, 30,000; MSn AGC Target, 50,000). Data files were analysed by Proteome Discoverer 2.0 (www.thermoscientific.com), using Mascot 2.4.1 (www.matrixscience.com) and the SwissProt Human database. Mascot result files were examined using Scaffold Q/Q+S V4.4.7 (www.proteomesoftware.com). Allowance was made for the following fixed (carbamidomethyl [C]) and variable modifications (oxidation [M], deoxidation [M] and Farnesyl [C]). Error tolerances were 10 parts per million for MS1 and 0.6 kD for MS2.

## Data Availability

The raw MS proteomics data have been deposited to the ProteomeXchange Consortium (http://proteomecentral.proteomexchange.org) through the PRIDE (PRoteomics IDEntifications) partner repository with the data set identifier PXD023121. This is an updated analysis to search for farnesyl (Cys) modification on FAM83F from a previously performed MS data (PXD009335) (1).

## Supplementary Information

## Acknowledgements

We thank E Allen, L Fin, J Stark, and A Muir for help and assistance with tissue culture, the staff at the DNA sequencing services (School of Life Sciences, University of Dundee), the cloning, antibody and protein production teams within the Medical Research Council-Protein Phosphorylation and Ubiquitylation Unit (MRC-PPU) reagents and services (University of Dundee), coordinated by J Hastie. We thank the staff at the Dundee Imaging Facility (School of Life Sciences, University of Dundee) and the Flow Cytometry Facility (School of Life Sciences, University of Dundee) for their invaluable help and advice throughout this project. We thank the Crick Aquatics Team (Biological Research Facility Science Technology Park) for *Xenoper* care and husbandry and the Crick Genomics Equipment Park for Sanger sequencing services. We thank all members of the Sapkota and Smith labs for their highly appreciated experimental advice and/or discussions. Funding: K Dunbar is supported by an MRC Career Development Fellowship. GP Sapkota is supported by the UK Medical Research Council (grants MC_UU_00018/6 and MC_UU_12016/3) and the pharmaceutical companies supporting the DSTT (Boehringer-Ingelheim, GlaxoSmithKline, Merck-Serono). RA Jones, K Dingwell and JC Smith are supported by the Francis Crick Institute, which receives its core funding from Cancer Research UK (FC001-157), the UK MRC (FC001-157), and the Wellcome Trust (FC001-157).

## Author Contributions

K Dunbar: data curation, formal analysis, investigation, and writing—original draft, review, and editing.
RA Jones: data curation, formal analysis, investigation, and writing—review and editing.
K Dingwell: investigation.
TJ Macartney: CRISPR strategy, design, and methodology as well as cloning.
JC Smith: formal analysis, supervision, funding acquisition, project administration, and writing—review and editing.
GP Sapkota: conceptualization, formal analysis, supervision, funding acquisition, project administration, and writing—review and editing.

### Conflict of Interest Statement

The authors declare that they have no conflict of interest.

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
