## [Reviewer comments · Life Science Alliance]

Life Science Alliance

FAM83F regulates canonical Wnt signalling through an interaction with CK1 α

Karen Dunbar, Rebecca Jones, Kevin Dingwell, Thomas Macartney, James Smith, and Gopal Sapkota

DOI: <https://doi.org/10.26508/lsa.202000805>

Corresponding author(s): Gopal Sapkota, University of Dundee

Review Timeline:	Submission Date:	2020-06-01
	Editorial Decision:	2020-07-31
	Revision Received:	2020-10-30
	Editorial Decision:	2020-12-08
	Revision Received:	2020-12-14
	Accepted:	2020-12-15

Scientific Editor: Shachi Bhatt

Transaction Report:

July 31, 2020

Re: Life Science Alliance manuscript #LSA-2020-00805-T

Dr. Gopal P. Sapkota
University of Dundee
MRC Protein Phosphorylation and Ubiquitylation Unit
School of Life Sciences
Dow Street
Dundee, Scotland DD1 5EH
United Kingdom

Dear Dr. Sapkota,

Thank you for submitting your manuscript entitled "FAM83F regulates canonical Wnt signalling through an interaction with CK1 α " to Life Science Alliance, which was now assessed by three referees, whose reports are copied below.

The referees appreciate the analysis, but they also raise significant concerns that need to be addressed for publication in this journal. For example, referees require

- more insight into the hierarchy between FAM83F and FAM83G in regulation of Wnt.
- the effect of FAM83F knockdown on axis duplication in *Xenopus*.

Given these positive recommendations, we would like to invite you to revise your manuscript with the understanding that the referee concerns (as in their reports) must be fully addressed and their suggestions taken on board. Please address all referee concerns in a complete point-by-point response. Acceptance of the manuscript will depend on a positive outcome of a second round of review, and we require strong support from referees for publication here.

In our view these revisions should typically be achievable in around 3 months. However, we are aware that many laboratories cannot function fully during the current COVID-19/SARS-CoV-2 pandemic and therefore encourage you to take the time necessary to revise the manuscript to the extent requested above. We will extend our 'scoping protection policy' to the full revision period required. If you do see another paper with related content published elsewhere, nonetheless contact me immediately so that we can discuss the best way to proceed.

Thank you for this interesting contribution to Life Science Alliance. We are looking forward to receiving your revised manuscript.

Sincerely,

Reilly Lorenz
Editorial Office Life Science Alliance
Meyerhofstr. 1
69117 Heidelberg, Germany
t +49 6221 8891 414
e contact@life-science-alliance.org
www.life-science-alliance.org

B. MANUSCRIPT ORGANIZATION AND FORMATTING:

Reviewer #1 (Comments to the Authors (Required)):

General comments:

Dunbar et al seek to investigate the role of FAM83F, a N-terminal domain containing protein which is part of the FAM83 family. This is a continuation of the work from their own laboratories, where they have painstakingly identified the role of this family of proteins; in particular, their interactions with and cellular sorting off the CK1 isoenzymes. Their over-arching and appealing hypothesis is that the FAM 83 family of proteins direct the CK1 isoforms with which they interact to distinct subcellular compartments there-by regulating their substrate pools. FM 83F which has a close homology to FM 83G was investigated for its role in modulating the Wnt pathway. Unlike FAM83F, the role of the FAM83G protein as a modulator of the Wnt pathway, is at least in part confirmed by the inherited genetic disease: Palmoplantar keratoderma.

In this manuscript the authors suggest that FAM83F also modulates the Wnt pathway but it's action upstream of the β catenin degradation complex unlike FAM83G which is downstream of this complex. Give some of the reservations I have about the data and interpretation of the findings (see below) one clear question that comes to mind is whether the phenotypes obtained by FAM83F knockdown or mutants that do not interact with be rescued by FAM83G.

Specific comments:

The paper was difficult to follow with reasons for undertaking experiments not clearly explained. In this context it would be useful if a figure was included to indicate the main domains of the FAM83F protein with the amino acid sequence numbered to show locations of the CK1a and CAAX binding domain etc for both xenopus and human proteins as well as the mRNAs used
It would also have been helpful to number the pages of the manuscript to enable easier review of this proposal.

What happens to axis duplication in Xenopus if FAM83F is knocked down?

Which of the mRNAs rescue this phenotype?

It is clear that the FAM83F F284/288A mutations will abrogate CK1a binding. What is the reason for this use of the D250A mutation? And how does it reduce the CK1a binding compared to wild type? in figure 2A/B, CK1a levels both in the nucleus and cytosol are clearly decreased. What happens to β catenin levels both in this context? Were the cells stained for β catenin? In the canonical Wnt pathway, CK1a is part of the complex that sequesters through degradation in the cytosol. Hence nuclear β catenin should increase Fig 2C suggests this might be the case vs GFP controls in the presence of Wnt3A conditioned media. I accept that the magnitude of increase is not as high as with FAM83F overexpression or with loss of membrane localisation. This however raises the question as to which mechanism is more physiologically relevant and how relevant are these observations in pathological states such as cancer? Are mutations in the FAM83F protein in the CAAX domain or CK1a binding domain present in cancer?

The explanation for the contradictory observations made between overexpression of the FAM83F c497A in xenopus and cell lines compared to monoallelic knock in of the same mutations is explained by high levels FAM83F c497A of redirection of endogenous CK1a to the nucleus and potentially reducing the cytosolic pool. This explanation is weak for the following reasons:

Data based on a single clone

Lack of a bi allelic knock-in

In Fig 6 A where FAM83F c497A is expressed there is no decrease in cytosolic CK1a compared to wild type

These issues need to be addressed.

It is important to explain the dose of the GSK3b inhibitor dose used as this is quite high (5mM).

What was the viability of the cells at this concentration? Was a dose response curve undertaken?

Reviewer #2 (Comments to the Authors (Required)):

Summary

Dunbar and co-workers follow up on an observation from the lab's 2018 Science Signaling paper, where they find that over-expression of FAM83F enhances Wnt/ β -catenin signaling in U2OS cells. Here they confirm and extend that finding, with evidence in xenopus and several human cancer cell lines that FAM83F interaction with CK1 α is required for this effect. The role of FAM83F appears to be upstream of the β -catenin destruction complex, but the site of action remains unresolved. The conclusion that membrane localization is required in colon cancer cells seems inconsistent with the data in U2OS cells in Fig 2C.

The experiments are well controlled, the manuscript is well written, and the conclusions are generally supported by the data. One weakness is the extensive analysis Wnt/ β -catenin signaling in cell lines with mutations in β -catenin and APC. Overall, the manuscript advances our understanding of the FAM83F protein function.

Major comments

The effect of the CaaX mutant appears different in U2OS and HCT116 cells. This should be discussed.

Minor comments

For the mutant analysis in Sup fig 2, a map/cartoon of the protein indicating the various domains and mutants would be helpful.

The font size for the superscripts in Sup 2A and B is too small for me to read unless markedly blown up.

Plunger plots as in Fig 2C, 4A, 5C/E, 6B/D should be replaced. Experiments with small samples sizes should use scatter-plots or similar methods that allow evaluation of the distribution of the data. c.f. Weissgerber et al. (2015) Beyond Bar and Line Graphs: Time for a New Data Presentation Paradigm. PLoS Biol 13(4): e1002128. doi:10.1371/journal.pbio.1002128

Reviewer #3 (Comments to the Authors (Required)):

This is a nice study about one of the members of the FAM38 family, that shows that FAM83F interacts with casein kinase 1 alpha. This interaction appears to sequester CK1alpha and, as a consequence, it has an activating effect on Wnt signalling, upstream of GSK3. It is proposed that

anchoring to the plasma membrane participates to the process of CK1alpha sequestration.

The conclusions are supported by clean high quality experimental data.

I have some comments and queries, but these are restricted to the text.

I would like also to comment on the careful interpretation of the data, reflecting a scholar spirit and critical thinking that one does not find as systematically as one would wish in this field.

Comments:

1) The authors conservatively propose that the impact of FAM83F on the Wnt pathway results from the balance between the different partners, in particular concerning the availability of CK1alpha. Thus, unlike what I have often seen in too many other studies, FAM83F is not presented as a direct component of the Wnt pathway". Rather, the expression of FMA38F indirectly influences the output of this pathway. It is nevertheless quite possible that in some cancers, this property has been "exploited" to deregulate Wnt- β catenin activity. This is one example of what I consider as a lucid view of cell signalling, which I highly appreciate in this manuscript.

In my opinion, this could be even more explicitly discussed in the text.

2) Positioning of the FAM83F-CK1alpha interaction along the Wnt pathway.

The authors observe that decreased Wnt signalling resulting from FAM83F KO can be rescued using a GSK3 inhibitor, which nicely demonstrates that the FAM83F-CK1alpha interaction must act upstream of GSK3. The authors state that FAM83F acts "upstream" of the destruction complex". I disagree: Although, as mentioned in the manuscript, CK1 isoforms play multiple roles along this pathway, the most obvious step which involves CK1alpha is priming of beta-catenin within the Axin-APC complex. If there is any experiment that one may have liked to see included, this would be to test whether FAM83F competes with Axin for CK1alpha binding. One can safely bet that it will.

In any case, this most likely scenario should be mentioned at least in the discussion. This would certainly further clarify the picture.

3) The authors use zebrafish FAM83Fa, which is not an issue since these proteins are quite conserved. However, they state that the second gene FAM83Fb, does not cause the same Wnt activation. I have real hard time to believe this, considering that duplicated genes in zebrafish have only very minor differences in their coding sequence (unless there is a difference within a key sequence responsible for membrane localization or CK1alpha binding???) I did not check, but I would bet that there are larger variations between fish, frog and human than between a and b isoforms. I see a more likely explanation in the blot of Fig.1C: FAM83b was expressed at way higher levels. As commented by the authors for another point, expression levels can have complex effect on the complex balance of these pathways. It seems quite likely that high FAM83 expression does something else, who knows, perhaps it ends up titrating another CK1 isoform, and/or titrates another CK1alpha pool that act antagonistically... This could be an avenue to explore in the future. For now, I would simply remove the FAM83Fb data.

3) page 6, top, and Fig.2B: "GFP-FAM83F is present predominately at the plasma membrane". I don't think that one can see predominant membrane staining in these images. Please rephrase.

Related: In fig.5, cell fractionation suggests that it is indeed enriched in the membrane fraction.

However, this would be valid only if what was loaded for each fraction reflect proportionality for the

three fractions. This is not indicated in the methods.

4) Page 7, top: "FAM68F... was undetectable in many other cell lines". This is interesting, perhaps the authors may include a comment in the discussion. Is there anything known about specific expression in normal tissues?

5) Page 8 and Fig.4B: Wnt activation should cause increased LRP6 phosphorylation and high total beta-catenin levels. Wnt3A-CM treatment does not show these effects on these signals. Perhaps LRP6 phosphorylation drops after 6hrs incubation. As for total beta-catenin, the increase in the "free total pool" is probably masked by high levels associated with cadherins. There should be a short comment to this lack of visible effect. As for "active beta-catenin", often these "specific antibodies" don't work as specifically as one may wish, thus I am not surprised.

Responses to reviewer's comments: The reviewer's comments are *italicised*, and our responses appear as non-italicised fonts. New data and figures are indicated with bold face fonts.

LSA-2020-00805-T: FAM83F regulates canonical Wnt signalling through an interaction with CK1 α

Reviewer #1

General comments:

Dunbar et al seek to investigate the role of FAM83F, a N-terminal domain containing protein which is part of the FAM83 family. This is a continuation of the work from their own laboratories, where they have painstakingly identified the role of this family of proteins; in particular, their interactions with and cellular sorting off the CK1 isoenzymes. Their overarching and appealing hypothesis is that the FAM83 family of proteins direct the CK1 isoforms with which they interact to distinct subcellular compartments there-by regulating their substrate pools. FM 83F which has a close homology to FAM83G was investigated for its role in modulating the Wnt pathway. Unlike FAM83F, the role of the FAM83G protein as a modulator of the Wnt pathway, is at least in part confirmed by the inherited genetic disease: Palmoplantar keratoderma.

In this manuscript the authors suggest that FAM83F also modulates the Wnt pathway but it's action upstream of the β catenin degradation complex unlike FAM83G which is downstream of this complex. Give some of the reservations I have about the data and interpretation of the findings (see below) one clear question that comes to mind is whether the phenotypes obtained by FAM83F knockdown or mutants that do not interact with be rescued by FAM83G.

Response: We thank the reviewer for an in-depth and constructive appraisal of our manuscript. The reviewer is correct to highlight the potential interplay of FAM83F and FAM83G proteins within the Wnt signalling pathway. To address whether FAM83G can restore the Wnt deficit in FAM83F^{-/-} cells, we transiently expressed GFP-FAM83G in HCT116 wild-type and FAM83F^{-/-} cells and evaluated the Wnt signalling output as determined by *Axin2* transcript expression (**Response Figure 1**). As previously shown in the manuscript, non-transfected FAM83F^{-/-} cells had significantly reduced *Axin2* expression following Wnt3A-CM treatment compared to wild-type cells (Figure 4A). The transient expression of GFP-FAM83G protein, which was confirmed by western blotting, did not increase *Axin2* expression in FAM83F^{-/-} HCT116 cells following Wnt3A-CM treatment (**Response Figure 1**). Therefore, our data indicates that FAM83G cannot rescue the Wnt signalling deficit observed in FAM83F^{-/-} cells, suggesting that the role of FAM83F in Wnt signalling perhaps occurs upstream of that of FAM83G. Consistent with this notion, we present evidence that FAM83F acts upstream of GSK-3 β , and we previously showed that FAM83G acts downstream of GSK-3 β in the Wnt signalling pathway (1). The precise interplay between FAM83F and FAM83G in Wnt signalling is an important unanswered question, which we are hoping to address in the future by dissecting the Wnt3A-induced phospho-proteomic landscape in wild-type, FAM83F^{-/-} and FAM83G^{-/-} cells to identify key CK1 α substrates that are dependent on FAM83F and FAM83G.

Response Figure 1: Transient FAM83G expression cannot rescue the Wnt signalling deficit in FAM83F^{-/-} cells. (A) qRT-PCR was performed using cDNA from HCT116 wild-type and FAM83F^{-/-} (clone.1) cells transiently expressing GFP-FAM83G protein and following treatment with L-CM or Wnt3A-CM for 6 h, and primers for *Axin2* and *GAPDH* genes. *Axin2* mRNA expression was normalised to *GAPDH* mRNA expression and represented as fold change compared to L-CM treated cells. Data presented as scatter graph illustrating individual data points with an overlay of the mean \pm standard deviation. Statistical significance was determined by unpaired Student's t-test. Statistically significant p-values are denoted by asterisks (**** <0.0001, *** <0.001, ** <0.01, * <0.05). **(B)** Lysates of HCT116 wild-type and FAM83F^{-/-} (clone.1) cells transiently expressing GFP-FAM83G were separated by SDS-PAGE and subjected to western blotting with the indicated antibodies to confirm expression of GFP-FAM83G.

Specific comments:

The paper was difficult to follow with reasons for undertaking experiments not clearly explained. In this context it would be useful if a figure was included to indicate the main domains of the FAM83F protein with the amino acid sequence numbered to show locations of the CK1a and CAAX binding domain etc for both xenopus and human proteins as well as the mRNAs used.

Response: We thank the reviewer for the comments. We have adjusted the text to improve clarity of rationale. We have also added illustrations of the protein domains and all mutant proteins used in this study. Specifically, illustrations of zebrafish Fam83fa protein and mutants are included in **Figure 1A** and **Supplementary Figure 2A** with illustrations of human FAM83F protein and mutants included in **Figure 2A** and **Figure 6A**.

It would also have been helpful to number the pages of the manuscript to enable easier review of this proposal.

Response: We have added page numbers to the manuscript.

What happens to axis duplication in Xenopus if FAM83F is knocked down? Which of the mRNAs rescue this phenotype?

Response: The axis duplication phenotype observed in *Xenopus* is an overexpression phenotype and a response to increased canonical Wnt signalling (2). We employed this assay to screen the potential of FAM83 proteins to activate Wnt signalling. As for the reviewer's suggestion, we would not anticipate any axis phenotype in *Xenopus* after knockdown of FAM83F as no FAM83F knock-out cell lines have exhibited an increase in canonical Wnt signalling. Whilst we believe that an *in vivo* model of FAM83F knock-out would be an important next step in the evaluation of FAM83F biology, unfortunately that is beyond the remit of this project. We have adjusted the text regarding the introduction of the axis duplication model to the following: "The activation of the canonical Wnt signalling pathway by ectopic expression of Wnt ligands and mediators in early *Xenopus* embryos causes axis duplication (2). Previously, we showed that injection of *Xenopus* embryos with FAM83G mRNA into a ventral blastomere at the four-cell stage induced secondary axis formation (1). Ectopically expressing mRNA in early *Xenopus* embryos is thus an efficient method for screening potential regulators of canonical Wnt signalling." We apologise for the initial confusion and hope this aids the reader's understanding of the *Xenopus* axis duplication experiments.

It is clear that the FAM83F F284/288A mutations will abrogate CK1a binding. What is the reason for this use of the D250A mutation? And how does it reduce the CK1a binding compared to wild type?

Response: We apologise that our reasons behind the use of D250A were not clear in the original text. The D250, F284, F288 amino acids are highly conserved across all FAM83 proteins. Indeed, point mutation of D250 in FAM83F or the equivalent residues in other FAM83 proteins has been shown to reduce the interaction between FAM83 and CK1 proteins (1, 3). We have improved the introduction of this point mutant with the addition of the following text: "In addition to the previously described conserved F-X-X-X-F motif, which

is required for the FAM83-CK1 interactions, a separate conserved residue that maps to an aspartic acid at 250 in FAM83F was identified which when mutated to an alanine can also disrupt FAM83-CK1 interactions (3) (Fig. 2A).” We demonstrate by GFP immunoprecipitations that GFP-FAM83F^{D250A} protein has minimal interaction with CK1 α compared to wild type GFP-FAM83F as shown in Figure 2B. Demonstrating that multiple point mutants of FAM83F which abolish CK1 binding all cause similar phenotypes, serves as a robust evidence that FAM83-CK1 binding is essential for FAM83F biology.

In figure 2A/B, CK1a levels both in the nucleus and cytosol are clearly decreased. What happens to b catenin levels both in this context? Were the cells stained for b catenin? In the canonical Wnt pathway, CK1a is part of the complex that sequesters through degradation in the cytosol. Hence nuclear b catenin should increase Fig 2C suggests this might be the case vs GFP controls in the presence of Wnt3A conditioned media. I accept that the magnitude of increase is not as high as with FAM83F overexpression or with loss of membrane localisation.

Response: We thank the reviewer for suggesting this experiment and agree that the expression of nuclear β -catenin may help explain the increase in Wnt signalling induced by expression of GFP-FAM83F protein or point mutants. To address this issue, we completed β -catenin staining in U2OS Flp/Trx cells expressing GFP-only, GFP-FAM83F, GFP-FAM83F^{C497A}, GFP-FAM83F^{D250A} or GFP-FAM83F^{F284A/F288A} following L-CM or Wnt3A-CM treatments (**Supplementary Figure 4A**). In all cell lines, β -catenin staining was predominately at the plasma membrane, which is to be expected due to the high proportion of β -catenin protein located at the adherens junctions. Following Wnt3A-CM treatment, there is slight increase in nuclear β -catenin staining in all cell lines, but the changes in levels across cell lines and treatments are quite hard to visualise and infer, as only a small of proportion of β -catenin appears to be translocating to the nucleus. Therefore, we also performed a cytoplasmic/nuclear fractionation in these cell lines following L-CM or Wnt3A-CM treatment for 6 hours (**Supplementary Figure 4B**). Again, the increase in nuclear β -catenin following Wnt3A-CM treatment is apparent in all cell lines but there is no observable difference in nuclear β -catenin levels between cells expressing GFP-FAM83F or GFP-FAM83F^{C497A} compared to GFP. The dual luciferase assay exploits a TCF-binding site upstream of a luciferase gene with β -catenin binding to TCF to induce expression of luciferase. Thus, an increase in luciferase activity reflects an increase in “free” β -catenin but this perhaps does not necessarily require nuclear translocation of β -catenin, as the TCF-luciferase was transiently transfected in cells. Expression of GFP-FAM83F and GFP-FAM83F^{C497A} increases cytoplasmic β -catenin levels basally compared to cells expressing GFP only or GFP-FAM83F^{F284A/F288A}, as demonstrated by subcellular fractionation (**Supplementary Figure 5**). This increase in “free” β -catenin potentially explains the increased TCF-luciferase activity detected upon expression of GFP-FAM83F and GFP-FAM83F^{C497A}. We have added this data to the manuscript. As overexpression of proteins and transcriptional luciferase assays in general are often prone to experimental artefacts, these were principally employed as initial screens to assess a broad impact of FAM83F and mutants on Wnt responses and we feel that our data from FAM83F knockout and farnesyl-deficient mutant knock-in cells provide better insights into the role of FAM83F in Wnt signalling.

This however raises the question as to which mechanism is more physiologically relevant and how relevant are these observations in pathological states such as cancer? Are mutations in the FAM83F protein in the CAAX domain or CK1a binding domain present in cancer?

Response: This is the first report in which we link FAM83F to Wnt signalling. Thus far, we are not aware of any reports of pathogenic FAM83F mutations in any diseases, including cancer. Nonetheless, we analysed FAM83F mutations in cancer using data from the catalogue of somatic mutations in cancer (COSMIC) database (<https://cancer.sanger.ac.uk/cosmic>). A small proportion of samples (366 from 37,486 samples) contained mutations in the FAM83F protein, with most of these mutations resulting in single amino acid substitutions. A vast majority of these are low frequency mutations that are evenly spread out throughout the FAM83F protein, with no indication of any mutation hotspots (**Response Figure 2**), suggesting possible silent carrier mutations. There does not seem to be any tissue specificity with the mutations detected across a wide variety of tissues. There is only one reported substitution within either the CAAX or FXXXF motifs which is a E287K substitution occurring once in a melanoma sample. Therefore, there is no evidence that mutations within either the CAAX or FXXXF motifs are over-represented in cancer. However, there is growing evidence that FAM83F protein is overexpressed in several cancers including glioma (4), lung cancer (5), oesophageal cancer (6) and thyroid cancer (7). It remains unknown whether high FAM83F protein expression promotes oncogenesis in these cancers or if its expression is a consequence of the cell alterations which occur in oncogenesis such as an increase in cell proliferation, migration and invasion. We are hopeful that our study will open up opportunities for other groups to interrogate the exciting questions that the reviewer raises.

Response Figure 2: Mutational overview of FAM83F protein in cancer. Histogram of all observed amino acid substitutions in the FAM83F protein across 37,486 cancer samples from the COSMIC database. Height of each bar represents the number of samples containing each specific amino acid substitution.

The explanation for the contradictory observations made between overexpression of the FAM83F^{C497A} in xenopus and cell lines compared to monoallelic knock in of the same mutations is explained by high levels FAM83F^{C497A} of redirection of endogenous CK1α to the nucleus and potentially reducing the cytosolic pool. This explanation is weak for the following reasons:

Data based on a single clone

Lack of a bi allelic knock-in

In Fig 6 A where FAM83F^{C497A} is expressed there is no decrease in cytosolic CK1α compared to wild type

These issues need to be addressed.

Response: We thank the reviewer for these comments and wholly agree that our initial inability to obtain a homozygous FAM83F^{C497A} knock-in was a limitation. We repeated the CRISPR-Cas9 knock-in experiments in HCT116 cells, and successfully managed to isolate three FAM83F^{C497A} knock-in clones (clones 1-3) which were confirmed by sequencing as homozygous (clones 1&3) and heterozygous (clone 2; the second allele had additional random insertions thereby also predictive of disrupting FAM83F membrane localisation) for the FAM83F^{C497A} knock-in (**Supplementary Figure 10**). Subcellular fractionation of HCT116 wild-type and HCT116 FAM83F^{C497A} (cl.1-3) cell extracts show a significant reduction in FAM83F and CK1α protein levels in the membrane enriched fractions of HCT116 FAM83F^{C497A} cells compared to HCT116 wild-type cells (**Figure 6B&C**). CK1α IPs from HCT116 (wild-type, FAM83F^{-/-} (cl.1) and FAM83F^{C497A} (cl.1-3)) extracts illustrate that the interaction between FAM83F and CK1α is maintained in wild-type and FAM83F^{C497A} (cl.1-3) cell lines but not FAM83F^{-/-} (cl.1) cells (**Figure 6D**). To assess the effect on Wnt signalling, *Axin2* transcripts were measured in HCT116 wild-type and HCT116 FAM83F^{C497A} (cl.1-3) cells following treatment with L-CM or Wnt3A-CM for 6 h (**Figure 6E**). HCT116 FAM83F^{C497A} (cl.1) cells had significantly reduced *Axin2* expression following Wnt3A-CM compared to wild-type cells. HCT116 FAM83F^{C497A} (cl.2) and HCT116 FAM83F^{C497A} (cl.3) cells had significantly reduced *Axin2* expression following treatment with both L-CM and Wnt3A-CM compared to wild-type cells. These results confirm that farnesylation of FAM83F is required to mediate Wnt signalling in HCT116 cells.

The reviewer is correct to highlight that the reduction in Wnt signalling in HCT116 FAM83F^{C497A} cell lines contrasts the Wnt activation observed following GFP-FAM83F^{C497A} expression in U2OS cells. CK1α impacts Wnt signalling in both a positive and negative way and, under physiological conditions, its activity is tightly regulated. We hypothesize that the differences between overexpressed and endogenous FAM83F^{C497A} can be explained by the re-direction of CK1α with overexpressed FAM83F^{C497A} binding a higher proportion of CK1α, thereby disrupting its homeostatic regulation, compared to endogenous FAM83F protein and redirecting CK1α to the nucleus. Therefore, there is a large reduction in cytosolic CK1α, which, due to the removal of CK1α phosphorylation at the β-catenin destruction complex, increases the Wnt signalling output. Conversely, endogenous FAM83F^{C497A} binds a smaller proportion of CK1α compared to overexpressed protein, so although some CK1α is redirected to the nucleus we don't see the same reduction in cytosolic CK1α. In addition, endogenous FAM83F^{C497A} is mis-localised from the membrane which in turn re-directs CK1α away from the membrane which has an inhibitory effect on the Wnt signalling pathway. The inhibition of Wnt signalling through re-direction of CK1α from the membrane is not observed in overexpression cell lines because these cells still contain endogenous FAM83F protein.

To confirm these changes in CK1 α localisation experimentally, we undertook subcellular fractionation in U20S Flp/Trx cells (expressing GFP only, GFP-FAM83F and GFP-FAM83F^{C497A}) and observed a reduction in cytosolic CK1 α in cells expressing GFP-FAM83F^{C497A} compared to those expressing GFP control (**Supplementary Figure 11A**). We also observed a significant reduction in phosphorylation of β -catenin (Ser45), following treatment with L-CM or Wnt3A-CM for 6 h, in cells expressing GFP-FAM83F^{C497A} compared to GFP-FAM83F or GFP only (**Supplementary Figure 11B&C**). This indicates that expression of GFP-FAM83F^{C497A} removes CK1 α from the β -catenin destruction complex, which would explain the increased cytoplasmic β -catenin (**Supplementary Figure 5**) and increased luciferase activity (Figure 2D) we observe upon expression of GFP-FAM83F^{C497A}.

It is important to explain the dose of the GSK3b inhibitor dose used as this is quite high (5mM). What was the viability of the cells at this concentration? Was a dose response curve undertaken?

Response: We thank the reviewer for bringing this to our attention. We performed a dose response curve in HCT116 cells (**Response Figure 3**). We see almost as robust an increase in *Axin2* expression with 0.5 μ M CHIR99021 as 5 μ M. Thus, we have repeated the GSK-3 β inhibitor experiments using 0.5 μ M and have now replaced **Figure 5A** (HCT116 cells) and **Supplementary Figure 9C** (U20S cells).

A.

HCT116 wild-type *Axin2* transcript expression

B.

HCT116 wild-type

Response Figure 3: CHIR99021 dose response. (A) qRT-PCR was performed using cDNA from HCT116 wild-type cells following treatment with varying concentrations of CHIR99021 for 6 hours, and primers for *Axin2* and *GAPDH* genes. *Axin2* mRNA expression was normalised to *GAPDH* mRNA expression and represented as fold change compared to untreated cells. Data presented as scatter graph illustrating individual data points with an overlay of the mean \pm standard deviation. **(B)** Lysates of HCT116 wild-type cells treated with varying concentrations of CHIR99021 for 6 hours were separated by SDS-PAGE and subjected to western blotting with the indicated antibodies to confirm reduction in p-β-catenin (S33/S37/T41) following CHIR99021 treatment.

Reviewer #2

Summary

Dunbar and co-workers follow up on an observation from the lab's 2018 Science Signaling paper, where they find that over-expression of FAM83F enhances Wnt/ β -catenin signaling in U2OS cells. Here they confirm and extend that finding, with evidence in xenopus and several human cancer cell lines that FAM83F interaction with CK1 α is required for this effect. The role of FAM83F appears to be upstream of the β -catenin destruction complex, but the site of action remains unresolved. The conclusion that membrane localization is required in colon cancer cells seems inconsistent with the data in U2OS cells in Fig 2C.

The experiments are well controlled, the manuscript is well written, and the conclusions are generally supported by the data. One weakness is the extensive analysis Wnt/ β -catenin signalling in cell lines with mutations in β -catenin and APC. Overall, the manuscript advances our understanding of the FAM83F protein function.

Response: We thank the reviewer for a positive and constructive appraisal of our manuscript. We acknowledge that the predominate use of colorectal cancer cell lines which contain activating mutations of canonical Wnt signalling is a limitation of the project. The main reason for using these cell lines was the robust expression of FAM83F protein in these cells compared to the limited detection of FAM83F protein in other cell lines (Supplementary Figure 6). The low expression of FAM83F in other cell lines made it technically difficult to generate FAM83F^{-/-} or GFP/GFP FAM83F cell lines with CRISPR/Cas9 technology. We did create a U2OS FAM83F^{-/-} cell line which also shows a significant reduction in Wnt signalling (Supplementary Figure 9) but in these cells FAM83F protein was only detectable following immunoprecipitation, making these cells practically difficult to work with.

Major comments

The effect of the CaaX mutant appears different in U2OS and HCT116 cells. This should be discussed.

Response: We appreciate the comments from the reviewer. We have now addressed these comments with new experiments and detailed discussion. Please see full response to **reviewer 1**, who also raised the same concerns.

Minor comments

For the mutant analysis in Sup fig 2, a map/cartoon of the protein indicating the various domains and mutants would be helpful.

Response: We thank the reviewer, and other reviewers, for bringing this to our attention. We have added illustrations of the zebrafish Fam83fa protein domains and mutants used within this project in **Figure 1A** and **Supplementary Figure 2A**. We have also added illustrations of human FAM83F domains and point mutants used within this study in **Figure 2A** and **Figure 6A**.

The font size for the superscripts in Sup 2A and B is too small for me to read unless markedly blown up.

Response: We have increased the font size of all superscripts.

Plunger plots as in Fig 2C, 4A, 5C/E, 6B/D should be replaced. Experiments with small samples sizes should use scatter-plots or similar methods that allow evaluation of the distribution of the data. c.f. Weissgerber et al. (2015) Beyond Bar and Line Graphs: Time for a New Data Presentation Paradigm. PLoS Biol 13(4): e1002128. doi:10.1371/journal.pbio.1002128.

Response: We have replaced all plunger plots with scatter plots illustrating the individual data points with an overlay of the mean and standard deviation. Except for the *Xenopus* axis duplication experiments which have large sample sizes with the specific sample numbers noted on each graph.

Reviewer #3:

This is a nice study about one of the members of the FAM38 family, that shows that FAM83F interacts with casein kinase 1 alpha. This interaction appears to sequester CK1alpha and, as a consequence, it has an activating effect on Wnt signalling, upstream of GSK3. It is proposed that anchoring to the plasma membrane participates to the process of CK1alpha sequestration.

The conclusions are supported by clean high quality experimental data.

I have some comments and queries, but these are restricted to the text.

I would like also to comment on the careful interpretation of the data, reflecting a scholar spirit and critical thinking that one does not find as systematically as one would wish in this field.

Comments:

1) The authors conservatively propose that the impact of FAM83F on the Wnt pathway results from the balance between the different partners, in particular concerning the availability of CK1alpha. Thus, unlike what I have often seen in too many other studies, FAM83F is not presented as a direct component of the Wnt pathway". Rather, the expression of FMA38F indirectly influences the output of this pathway. It is nevertheless quite possible that in some cancers, this property has been "exploited" to deregulate Wnt-βcatenin activity. This is one example of what I consider as a lucid view of cell signalling, which I highly appreciate in this manuscript.

In my opinion, this could be even more explicitly discussed in the text.

Response: We thank the reviewer for a positive and constructive appraisal of our manuscript. We agree with the reviewer in that we don't think FAM83F is a direct component of the Wnt signalling pathway as HCT116 FAM83F^{-/-} cells still have functioning Wnt signalling but at reduced levels compared to wild-type cells. Instead, we think FAM83F can modulate Wnt signalling through regulation of CK1α localisation. We have adjusted our discussion to highlight this mediator role of FAM83F. Specifically, the following text has been added on page 12: "We propose that FAM83F's biological role is to mediate the localisation of a proportion of CK1α and thus, to facilitate phosphorylation of a subset of CK1α substrates. Therefore, FAM83F may have effects in multiple signalling pathways, beyond Wnt signalling and p53 activity, through determining CK1α localisation and its substrates. Whilst targeting CK1α therapeutically would have multiple unwanted consequences, the potential to inhibit specific pools of CK1α and potentially specific substrates through inhibition of the FAM83F-CK1α complex could make FAM83F an attractive therapeutic target in cancer.

2) Positioning of the FAM83F-CK1alpha interaction along the Wnt pathway. The authors observe that decreased Wnt signalling resulting from FAM83F KO can be rescued using a GSK3 inhibitor, which nicely demonstrates that the FAM83F-CK1alpha interaction must act upstream of GSK3. The authors state that FAM83F acts "upstream" of the destruction complex". I disagree: Although, as mentioned in the manuscript, CK1 isoforms play multiple roles along this pathway, the most obvious step which involves CK1alpha is priming of beta-catenin within the Axin-APC complex. If there is any experiment that one may have liked to see included, this would be to test whether FAM83F competes with Axin for CK1alpha binding. One can safely bet that it will. In any case, this most likely scenario should be mentioned at least in the discussion. This would certainly further clarify the picture.

Response: We thank the reviewer for highlighting an important point. We appreciate that "upstream of GSK3 β " does not necessarily equate to "upstream of the β -catenin destruction complex". Thus, we have altered language to note that our data indicates FAM83F acts upstream of GSK3 β rather than the β -catenin destruction complex. We agree the effect of FAM83F on the Axin-CK1 α interaction is an important question. We performed immunoprecipitations with CK1 α antibody in HCT116 wild-type and FAM83F^{-/-} cells following L-CM or Wnt3A-CM treatment (**Response Figure 4**). CK1 α interacts with Axin-1 protein in wild-type and FAM83F^{-/-} cells following both L-CM and Wnt3A-CM treatment. Densitometry of Axin-1 protein levels normalised to CK1 α in the IP elutes indicates there is no significant change in the proportion of Axin-1 bound to CK1 α in FAM83F^{-/-} cells compared to wild-type cells. Additionally, we have tested the phosphorylation of β -catenin at Serine 45 in HCT116 and DLD-1 FAM83F^{-/-} cell lines following L-CM or Wnt3A-CM treatment and observed no changes in β -catenin phosphorylation (Figure 4C). Therefore, we have no evidence that the FAM83F-CK1 α complex is involved or required for CK1 α mediated phosphorylation of β -catenin. Given the strong reduction in Wnt signalling in FAM83F^{C497A} cell lines caused by loss of FAM83F and CK1 α protein from the membrane, we think it is most likely that FAM83F-CK1 α substrates reside at the plasma membrane. We are undertaking an unbiased phospho-proteomic screen to identify FAM83F-CK1 α substrates in Wnt signalling as a next step, although these are clearly beyond the scope of this manuscript.

Response Figure 4: Effect of FAM83F protein abundance on the CK1α-Axin-1 interaction. (A) Lysates of HCT116 wild-type and FAM83F^{-/-} (clone.1) cells following treatment with either L-CM or Wnt3A-CM for 6 h were subjected to immunoprecipitation with CK1α antibody. Input lysates and CK1α IP elutes were separated by SDS-PAGE and subjected to western blotting with the indicated antibodies. (B) Densitometry of Axin-1 and CK1α protein levels in the CK1α IP elutes were performed and Axin-1 protein levels were normalised to CK1α levels. Quantification of Axin-1/CK1α is presented as a fold-change compared to wild-type cells treated with L-CM. Data presented as scatter graph illustrating individual data points with an overlay of the mean ± standard deviation.

3) The authors use zebrafish FAM83Fa, which is not an issue since these proteins are quite conserved. However, they state that the second gene FAM83Fb, does not cause the same Wnt activation. I have real hard time to believe this, considering that duplicated genes in zebrafish have only very minor differences in their coding sequence (unless there is a difference within a key sequence responsible for membrane localization or CK1alpha binding???) I did not check, but I would bet that there are larger variations between fish, frog and human than between a and b isoforms. I see a more likely explanation in the blot of Fig. 1C: FAM83b was expressed at way higher levels. As commented by the authors for another point, expression levels can have complex effect on the complex balance of these pathways. It seems quite likely that high FAM83 expression does something else, who knows, perhaps it ends up titrating another CK1 isoform, and/or titrates another CK1alpha pool that act antagonistically... This could be an avenue to explore in the future. For now, I would simply remove the FAM83Fb data.

Response: We understand the differences between zebrafish Fam83fa and Fam83fb is an important question and currently we do not fully understand the biology of either protein. The reviewer is correct to highlight the differences in expression between Fam83fa and Fam83fb which we concede adds an additional variable in evaluating the role of each protein in the *Xenopus* axis duplication model. Unfortunately, the Fam83fa protein expression was consistently lower than Fam83fb so we cannot confirm if high Fam83fa expression would still induce axis duplication or, as hypothesised by the reviewer, have other effects due to changes in the balance between Fam83 and CK1 protein levels. However, we do show “high” expression of Fam83fa truncation mutants in Figure 1, specifically Fam83fa^{1-300(DUF)}, which still induces axis duplication. Thus, we do not think that the expression levels are the reason for differences between Fam83fa and Fam83fb. We believe the differences between the Fam83fa and Fam83fb sequences is the determining factor in whether the protein can induce axis duplication in *Xenopus*. Importantly, Fam83fb does not contain the predicted farnesylation sequence or the F-X-X-X-F motif. Our original hypothesis, based on Fam83fa mutants, was that this lack of F-X-X-X-F motif may explain the lack of axis duplication by Fam83fb. However, restoration of the F-X-X-X-F motif in Fam83fb (Fam83fb^{Y275F}) does not induce axis duplication in *Xenopus* models indicating that this is not the single determining factor in Fam83fb axis duplication (**Response Figure 5**). We predict the combined loss of the predicted farnesylation and the F-X-X-X-F motifs are most likely responsible for the differences in axis duplication, but we are aware there may be other important structural alterations which we do not currently fully understand. Considering we cannot answer definitively why Fam83fb does not induce axis duplication, this isoform is absent from mammals, and following the reviewer’s advice, we have removed the Fam83fb data from the manuscript.

Response Figure 5: *Xenopus* axis duplication assay with *fam83fb* mRNA. Percentage of *Xenopus* embryos showing the phenotypes in (Figure 1A) following injection with HA tagged zebrafish *fam83fa*^{1-300aa}, *fam83fb*^{1-300aa} or *fam83fb*^{1-300aa(Y275F)} mRNA. Data represents three independent experiments. Bar graph represents mean + standard deviation. Statistical significance was determined by two-way ANOVA with Dunnett's post-hoc test to compare the percentage of embryos displaying a wild-type phenotype. ****p<0.0001

3) page 6, top, and Fig.2B: "GFP-FAM83F is present predominately at the plasma membrane". I don't think that one can see predominant membrane staining in these images. Please rephrase.

Related: In fig.5, cell fractionation suggests that it is indeed enriched in the membrane fraction. However, this would be valid only if what was loaded for each fraction reflect proportionality for the three fractions. This is not indicated in the methods.

Response: We accept the reviewer's comment and we have rephrased the description of GFP-FAM83F immunofluorescence staining to the following: "Fluorescence microscopy shows that GFP-FAM83F is expressed at both the plasma membrane and nucleus in U2OS Flp/Trx cells (Fig. 2C)." Regarding the subcellular fractionation, equal protein concentrations were loaded for each subcellular fraction rather than specific proportions. We have added new text to the Materials & Methods section to clarify: "For subcellular fractions, 15 µg of protein was loaded for each fraction and the corresponding compartment-specific loading control was used for densitometry normalisation." We are aware that by loading equal

protein rather than adjusting each compartment to reflect proportion, we may over emphasise the amount of protein located in certain subcellular compartments. However, we have also demonstrated the membrane localisation of endogenous FAM83F by immunofluorescence in HCT116 cells (Figure 3E&F).

4) Page 7, top: "FAM68F... was undetectable in many other cell lines". This is interesting, perhaps the authors may include a comment in the discussion. Is there anything known about specific expression in normal tissues?

Response: We agree with the reviewer that low FAM83F protein abundance in most tested cell lines is an interesting point and we do not fully understand why the abundance is so low in these cells. We have tested abundance in mouse tissues and determined that the highest abundance was observed in the gastrointestinal tissues, where Wnt signalling plays a critical role (Figure 3A). This mirrors our cell line data (Supplementary Figure 6), in which colorectal cancer cell lines have the highest FAM83F protein abundance and thus we hypothesise that this may be related to the reliance of these cells and tissues on WNT signalling for homeostasis and maintenance. We have included a comment on FAM83F abundance in the discussion section as advised: "Little is known about the function of FAM83F and in most tissues and cell lines, except for the gastrointestinal tissues and colorectal cancer cell lines, FAM83F protein levels are undetectable. Interestingly, gastrointestinal tissues and colorectal cancer cells are reliant on canonical Wnt signalling for homeostasis and maintenance."

5) Page 8 and Fig.4B: Wnt activation should cause increased LRP6 phosphorylation and high total beta-catenin levels. Wnt3A-CM treatment does not show these effects on these signals. Perhaps LRP6 phosphorylation drops after 6hrs incubation. As for total beta-catenin, the increase in the "free total pool" is probably masked by high levels associated with cadherins. There should be a short comment to this lack of visible effect. As for "active beta-catenin", often these "specific antibodies" don't work as specifically as one may wish, thus I am not surprised

Response: We appreciate the reviewer's advice and agree that the lack of a visible activation of the Wnt signalling pathway, specifically LRP6 phosphorylation, upon Wnt3A-CM treatment is an important point. Whilst we can consistently observe an increase in *Axin2* transcript levels following Wnt3A-CM treatment, the changes in Wnt signalling pathway proteins which are often reported to be modified following Wnt3A activation, such as LRP6 phosphorylation and increased active β -catenin, are not always apparent in our hands – this despite asking for advice and obtaining reagents from leading Wnt signalling labs. On the reviewer's advice, we have added a comment to the results section noting the absence of visible changes in Wnt signalling pathway proteins upon Wnt3A stimulation. We have added the following text to page 8: "However, the absence of phosphorylation of LRP6 and increase in active β -catenin following Wnt3A stimulus is surprising and may reflect the transient nature of phosphorylation within signalling pathways, or a poor response to Wnt3A stimulus due to constitutively active Wnt signalling in these cell lines."

1. Bozatzí P, Dingwell KS, Wu KZ, Cooper F, Cummins TD, Hutchinson LD, et al. PAWS1 controls Wnt signalling through association with casein kinase 1 α . *EMBO Rep.* 2018;19(4).
2. Kühí M, Pandur P. Dorsal axis duplication as a functional readout for Wnt activity. *Methods Mol Biol.* 2008;469:467-76.
3. Fulcher LJ, Bozatzí P, Tachie-Menson T, Wu KZL, Cummins TD, Bufton JC, et al. The DUF1669 domain of FAM83 family proteins anchor casein kinase 1 isoforms. *Sci Signal.* 2018;11(531).
4. Xu L, Yu QW, Fang SQ, Zheng YK, Qi JC. MiR-650 inhibits the progression of glioma by targeting FAM83F. *Eur Rev Med Pharmacol Sci.* 2018;22(23):8391-8.
5. Fan G, Xu P, Tu P. MiR-1827 functions as a tumor suppressor in lung adenocarcinoma by targeting MYC and FAM83F. *J Cell Biochem.* 2019.
6. Mao Y, Liu J, Zhang D, Li B. miR-143 inhibits tumor progression by targeting FAM83F in esophageal squamous cell carcinoma. *Tumour Biol.* 2016;37(7):9009-22.
7. Fuziwara CS, Saito KC, Leoni SG, Waitzberg Â, Kimura ET. The Highly Expressed FAM83F Protein in Papillary Thyroid Cancer Exerts a Pro-Oncogenic Role in Thyroid Follicular Cells. *Front Endocrinol (Lausanne).* 2019;10:134.

December 8, 2020

RE: Life Science Alliance Manuscript #LSA-2020-00805-TR

Dr. Gopal P. Sapkota
University of Dundee
MRC Protein Phosphorylation and Ubiquitylation Unit
School of Life Sciences
Dow Street
Dundee, Scotland DD1 5EH
United Kingdom

Dear Dr. Sapkota,

Thank you for submitting your revised manuscript entitled "FAM83F regulates canonical Wnt signalling through an interaction with CK1 α ". We would be happy to publish your paper in Life Science Alliance pending final revisions necessary to meet our formatting guidelines.

Along with the points listed below, please also attend to the following,

- please merge the supplemental materials and methods section with the main manuscript materials and methods section
- please add callouts for Figure S3A, S3B, S3C, S4A, S4B, S7A, S7B, S10A, S10B
- please rename all supplemental figures as Figure S1 etc
- please remove panel label A from figures that only have one panel, eg. supplemental figure 1
- please reformat the references to 10 authors et al format

A. FINAL FILES:

B. MANUSCRIPT ORGANIZATION AND FORMATTING:

Sincerely,

Shachi Bhatt, Ph.D.
Executive Editor
Life Science Alliance
<https://www.lsjournal.org/>
Tweet @SciBhatt @LSAJournal

Reviewer #1 (Comments to the Authors (Required)):

The revised document is thorough with additional experiments now better explaining complex biology.

Reviewer #2 (Comments to the Authors (Required)):

The authors have satisfactorily responded to my comments.

Reviewer #3 (Comments to the Authors (Required)):

The authors have satisfactorily answered all queries. The revised manuscript presents a high quality study of an interesting regulatory mechanism.

December 15, 2020

RE: Life Science Alliance Manuscript #LSA-2020-00805-TRR

Dr. Gopal P. Sapkota
University of Dundee
MRC Protein Phosphorylation and Ubiquitylation Unit
School of Life Sciences
Dow Street
Dundee, Scotland DD1 5EH
United Kingdom

Dear Dr. Sapkota,

Thank you for submitting your Research Article entitled "FAM83F regulates canonical Wnt signalling through an interaction with CK1 α ". It is a pleasure to let you know that your manuscript is now accepted for publication in Life Science Alliance. Congratulations on this interesting work.

DISTRIBUTION OF MATERIALS:

Again, congratulations on a very nice paper. I hope you found the review process to be constructive and are pleased with how the manuscript was handled editorially. We look forward to future exciting submissions from your lab.

Sincerely,

Shachi Bhatt, Ph.D.

Executive Editor

Life Science Alliance

<https://www.lsjournal.org/>
